# Resistance of melanoma to immune checkpoint inhibitors is overcome by targeting the sphingosine kinase-1

Caroline Imbert [1,2,3], Anne Montfort[1,2,3], Marine Fraisse[1,2,3], Elie Marcheteau[1,2,3], Julia Gilhodes[4,5],
Elodie Martin[4,5], Florie Bertrand[1,2,3], Marlène Marcellin[3,6], Odile Burlet-Schiltz[3,6], Anne Gonzalez de Peredo[3,6],
Virginie Garcia [1,2,3], Stéphane Carpentier[1,2,3], Sophie Tartare-Deckert [7], Pierre Brousset[1,3,4],
Philippe Rochaix [1,5], Florent Puisset[1,3,4], Thomas Filleron[4,5], Nicolas Meyer[1,3,4], Laurence Lamant[1,3,4],
Thierry Levade[1,2,3,8], Bruno Ségui [1,2,3], Nathalie Andrieu-Abadie [1,2,3,9] & Céline Colacios [1,2,3,9]*

Immune checkpoint inhibitors (ICIs) have dramatically modified the prognosis of several advanced cancers, however many patients still do not respond to treatment. Optimal results might be obtained by targeting cancer cell metabolism to modulate the immunosuppressive tumor microenvironment. Here, we identify sphingosine kinase-1 (SK1) as a key regulator of anti-tumor immunity. Increased expression of SK1 in tumor cells is significantly associated with shorter survival in metastatic melanoma patients treated with anti-PD-1. Targeting SK1 markedly enhances the responses to ICI in murine models of melanoma, breast and colon cancer. Mechanistically, SK1 silencing decreases the expression of various immunosuppressive factors in the tumor microenvironment to limit regulatory T cell (Treg) infiltration. Accordingly, a SK1-dependent immunosuppressive signature is also observed in human melanoma biopsies. Altogether, this study identifies SK1 as a checkpoint lipid kinase that could be targeted to enhance immunotherapy.

[1] Inserm UMR1037, Cancer Research Center of Toulouse, 31037 Toulouse, France. [2] Equipe Labellisée Ligue Contre Le Cancer, 31037 Toulouse, France.
[3] Université Toulouse III—Paul Sabatier, 31062 Toulouse, France. [4] Institut Universitaire du Cancer Toulouse, Oncopôle, 31059 Toulouse, France. [5] Institut
Claudius Regaud, 31300 Toulouse, France. [6] Centre National de la Recherche Scientifique, Institut de Pharmacologie et de Biologie Structurale, 31077
Toulouse, France. [7] Centre Méditerranéen de Médecine Moléculaire, Inserm U1065, 06204 Nice, France. [8] Laboratoire de Biochimie, Institut Fédératif de
Biologie, CHU Purpan, 31059 Toulouse, France. [9] These authors contributed equally: Nathalie Andrieu-Abadie, Céline Colacios *email: celine.colacios@inserm.fr

The recent success of immunotherapy in melanoma relates to the use of monoclonal antibodies directed against the immune checkpoints such as programmed cell death protein-1 (anti-PD-1; pembrolizumab, nivolumab) and cytotoxic T-lymphocyte-associated protein-4 (anti-CTLA-4; ipilimumab). These immune checkpoint inhibitors (ICI) have demonstrated significant clinical efficacy in metastatic melanoma by reversing effector T-cell dysfunction and exhaustion, thereby enhancing their anti-tumoral properties[1]. However, a proportion of 40–60% of patients do not achieve any significant therapeutic response and a substantial proportion of responder experience tumor relapse within 2 years[2–5]. Consequently, innovative strategies and synergistic combination therapies, which block more than one immunomodulatory pathway are still needed to eradicate advanced melanomas[6]. A promising strategy is to modulate cancer cell metabolism, for instance sphingolipid metabolism, by targeting key enzymes involved in the production of oncometabolites, such as sphingosine-1-phosphate (S1P)[7].

S1P is produced by sphingosine kinases (SK) that catalyze the phosphorylation of sphingosine to S1P. The SK type 1 isoform (SK1), encoded by the *SPHK1* gene, which is overexpressed in numerous human tumors, including melanoma, leads to increased levels of S1P[8,9]. The SK1/S1P axis could modulate different hallmarks of cancer such as cell proliferation, cell death, metastasis and angiogenesis[10,11]. Moreover, S1P is a well-known regulator of lymphocyte trafficking and differentiation under different pathophysiological conditions[12,13]. However, the impact of increased SK1 expression in melanoma cells on the abundance, the functions and the phenotype of tumor-infiltrating lymphocytes (TILs) is unknown. TILs are a heterogeneous population for which frequency, localization, and subset ratio in solid tumors correlate with prognosis and immunotherapeutic responses[14,15]. CD8 + T cells play a central role in anti-tumor immunity whereas accumulation of Foxp3 + regulatory T cells (Treg) dampens effector function. Consequently, the CD8/Treg ratio in the tumor microenvironment (TME) constitutes a critical factor in immunotherapy[16,17]. How tumor cell metabolism, particularly sphingolipid metabolism, modulates this ratio needs further attention.

Here, we observe that high expression of SK1 in tumor cells is associated with shorter survival in melanoma patients treated with anti-PD-1. Interestingly, silencing of SK1 in preclinical models leads to attenuated tumor growth and Treg recruitment, and enhances the CD8/Treg ratio in tumors. Moreover, using epigenetic and pharmacological approaches to target SK1, we show that SK1 expression in melanoma impairs the responses to ICI. Our results demonstrate, that combining ICI and SK1 antagonism may represent the basis for innovative anti-melanoma therapies.

## Results

### SPHK1 expression inversely correlates with survival after ICI therapy. 
Analysis of two different cohorts from the Oncomine database indicated that *SPHK1* (encoding SK1) transcript levels were higher in human primary melanomas as compared to nevi (Fig. 1a, left panel); *SPHK1* expression was further increased in metastatic melanomas (Fig. 1a, right panel), suggesting that *SPHK1* expression might be associated with melanoma progression.

In order to evaluate whether *SPHK1* expression was related to the therapeutic outcome in advanced melanoma patients receiving anti-PD-1 therapy (Table 1), we analyzed *SPHK1* messenger RNA (mRNA) expression in tumor biopsies by in situ hybridization using the RNAscope technology.

Patients were separated in two groups according to the proportion of tumor cells positive for *SPHK1* mRNA (Low *SPHK1*: ≤ 50% tumor cells positive and High *SPHK1*: > 50% tumor cells positive) (Fig. 1b). Figure 1c shows representative

*SPHK1* staining for these two groups. Patients with low *SPHK1* expression had significantly longer progression-free survival and overall survival than those with high *SPHK1* expression ($p = 0.0112$ and $p = 0.0445$, respectively; log-rank test) (Fig. 1d, e), and patients with high *SPHK1* expression mostly failed to respond to anti-PD-1 therapy. These findings support the hypothesis that *SPHK1* expression represents a potential biomarker to predict tumor progression and resistance to anti-PD-1 in metastatic melanoma patients.

### SK1 silencing enhances anti-tumor immune response. 
In order to assess the impact of SK1 expression on melanoma growth, we generated stable SK1 knockdown cells using Yumm 1.7 cells derived from spontaneous murine melanoma driven by *Braf* activation, as well as *Pten* and *Cdkn2a* inactivation[18,19]. This cell line has previously been shown to resist PD-1 blockade[20]. The puromycin-resistant cell lines shSK1(1), shSK1(2) and shSK1(3), silenced for SK1, were obtained with three different shRNA sequences. shSK1 cells exhibited a markedly reduced enzymatic activity of SK1 (Fig. 2a and Supplementary Fig. 2a). While SK1 silencing did not modify in vitro Yumm cell proliferation (Fig. 2b), we observed a significant and sustained reduction in tumor growth after intradermal injection in C57BL/6 wild-type (WT) mice (Fig. 2c and Supplementary Fig. 2a). Interestingly, tumor regression was observed after day 12 for SK1-silenced melanoma cells, suggesting an increased anti-melanoma immune response. In addition, SK1 silencing did not alter Yumm tumor growth in immunodeficient NSG (Fig. 2d). In vivo depletion experiments demonstrated that CD4 and CD8 cells, but not NK cells, were important in the inhibition of tumor growth triggered by SK1 silencing (Supplementary Fig. 2b).

To investigate the impact of SK1 silencing in melanoma cells on the anti-tumor immune response, we analyzed the lymphocyte infiltration of tumors in mice injected either with shCtrl or shSK1(1) Yumm cells at day 11. Interestingly, SK1 silencing increased the proportion of CD8 + T cells while decreasing that of Foxp3 + CD4 + T cells (Treg), leading to an increased CD8/Treg ratio in shSK1 tumors as compared to shCtrl tumors-injected mice (Fig. 2e, f). Analysis of TIL proliferation, evaluated by Ki67 expression, showed that SK1 silencing markedly increased the percentage of Ki67 + CD8 + T cells and decreased that of Ki67 + Treg (Fig. 3a, b). Similar results were observed in tumor draining lymph nodes (TDLN), but not in non-TDLN (Supplementary Fig. 3).

Tumor SK1 downregulation increased the proportion of IFN-γ-producing CD8 + TILs (Fig. 3c) and CD107a + and granzyme + CD8 T cells (Supplementary Fig. 1c), as well as the expression of PD-1, CTLA-4, and TIM-3 on CD8 + TILs (Fig. 3d, e) that likely reflect an accumulation of activated CD8 + T cells at day 11. Indeed, these three checkpoint molecules are upregulated upon T-cell activation during the anti-tumor response, and tumor-infiltrating CTLA-4 + PD-1 + CD8 + T cells have been shown to contain the majority of tumor-antigen-specific T cells[21,22]. This phenomenon was accompanied by a significant decrease of CTLA-4 expression in Treg (Fig. 3f, g). These results strongly suggest that SK1 downregulation attenuated Treg accumulation into tumors, leading to an enhanced CD8 + T-cell activation.

To understand how tumor SK1 expression and Treg infiltration were related, the expression of immunosuppressive cytokines (TGF-β and IL-10) and chemokines (CCL17 and CCL22) that support Treg recruitment[23,24], was evaluated in tumors from mice injected with shCtrl or shSK1(1) Yumm cells. Our data show that SK1 silencing significantly reduced the expression of Foxp3, TGF-β, IL-10, CCL17, and CCL22 (Fig. 3h) without affecting that of IDO1, which exerts immunosuppressive effects

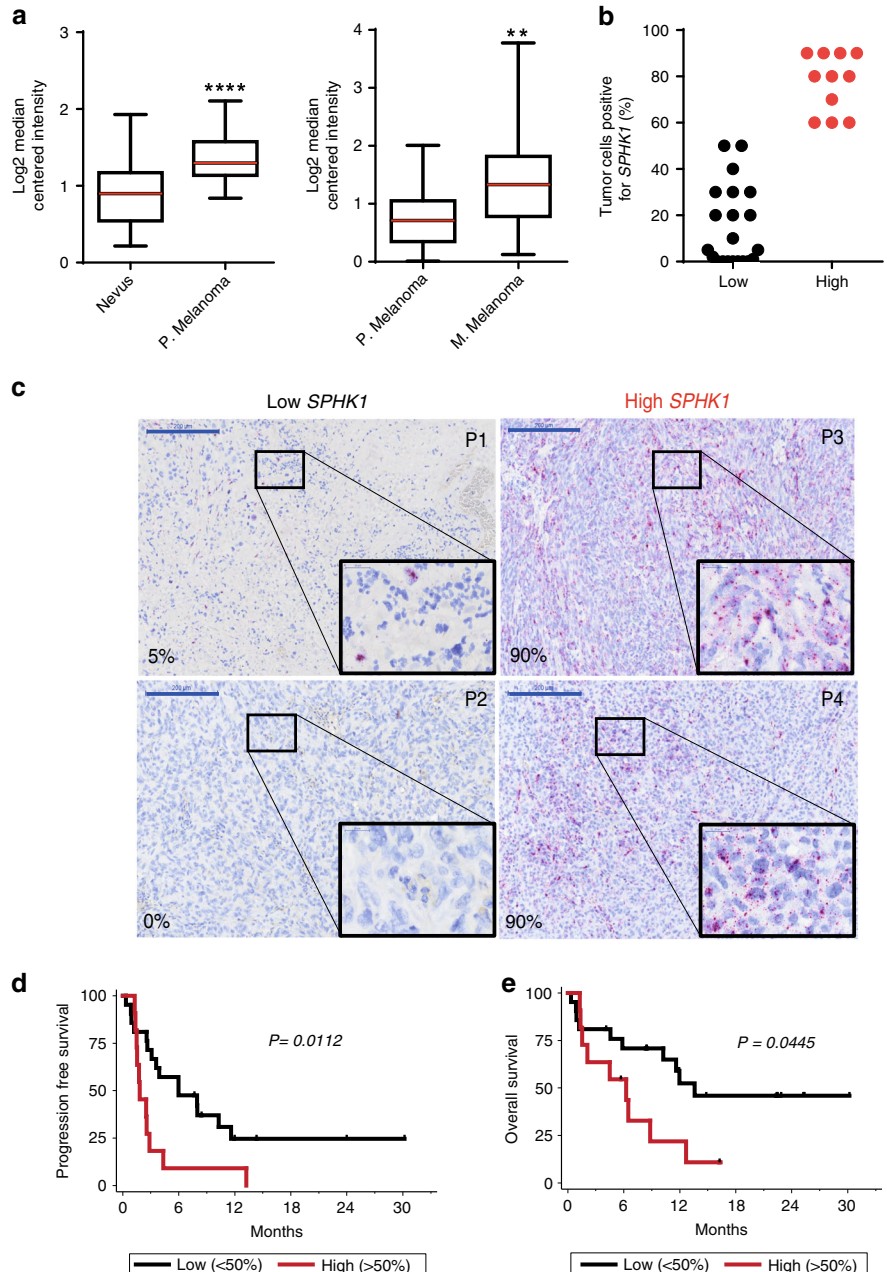

**Fig. 1 SPHK1 expression inversely correlates with survival after ICI therapy. a** *SPHK1* expression in human nevi ($n = 17$) compared to primary (P) melanoma ($n = 45$), and in primary melanoma ($n = 25$) compared to metastatic (M) melanoma ($n = 44$) based on the Oncomine database. Data were compared using Mann–Whitney test. The bottom-most and topmost horizontal lines, the lower and upper hinges, and the middle line of the boxplots indicate the minimum and maximum values, the 25th and 75th percentiles, and the median, respectively. **b** Percentages of cancer cells positive for *SPHK1* mRNA staining in metastatic melanoma tissues of 32 patients prior anti-PD-1 treatment (Low: $\leq 50\%$ of tumor cells are positive (black points); High: > 50% of tumor cells are negative (red points)). **c** Representative mRNA staining of low and high *SPHK1* expression. Skin (P1,P3) or lymph node (P2,P4) biopsies from patients (P). Percentages (%) indicate the proportion of cancer cells positive for *SPHK1* mRNA staining. Large and small blue lines represent 200 and 20 μm, respectively. **d** Progression-free survival and **e** overall survival curves of patients with >50% of melanoma cells positive for *SPHK1* (red line; $n = 11$) or <50% (black line; $n = 21$). Survival times were calculated from the first day of the cycle of anti-PD-1 post biopsy. Statistical significance was determined by log-rank test.

on T-cell proliferation. Similar results were obtained using shSK1 (2) melanoma cells (Supplementary Fig. 4), confirming that SK1 acts as a key driver of Treg-associated cytokine and chemokine expression.

**Downregulation of SK1 improves the efficacy of ICI.** Given that SK1 silencing led to an increased expression of immune checkpoint molecules on CD8 + T cells and to a reduced accumulation of Treg, we hypothesized that SK1 downregulation may improve the efficacy of ICI (e.g., anti-CTLA-4 and anti-PD-1 therapy). As illustrated in Fig. 4, whereas anti-CTLA-4 or anti-PD-1 monotherapy had limited effects (18% of tumor rejection) on established shCtrl tumors, SK1 silencing dramatically enhanced the response to anti-CTLA-4 or anti-PD-1 treatment, leading to tumor rejection in 100% and 67% of mice, respectively

## Table 1 Patient demographic and clinical characteristics.

| | Total N = 32 | Low SPHK1 (≤50%) N = 21 | High SPHK1 (>50%) N = 11 |
|---|---|---|---|
| Gender (n = 32) | | | p = 0.4424 |
| Male | 21 (65.6%) | 15 (71.4%) | 6 (54.5%) |
| Female | 11 (34.4%) | 6 (28.6%) | 5 (45.5%) |
| Age at treatment initiation (n = 32) | | | p = 0.7208 |
| ≤65 years | 13 (40.6%) | 8 (38.1%) | 5 (45.5%) |
| >65 years | 19 (59.4%) | 13 (61.9%) | 6 (54.5%) |
| Who performance status (n = 31) | | | p = 1.0000 |
| 0 | 20 (64.5%) | 13 (65.0%) | 7 (63.6%) |
| 1 | 11 (35.5%) | 7 (35.0%) | 4 (36.4%) |
| Missing | 1 | 1 | 0 |
| Stage (n = 32) | | | p = 0.1378 |
| IIIc | 5 (15.6%) | 5 (23.8%) | 0 (0.0%) |
| IV | 5 (15.6%) | 4 (19.0%) | 1 (9.1%) |
| IVa | 5 (15.6%) | 4 (19.0%) | 1 (9.1%) |
| IVb | 5 (15.6%) | 1 (4.8%) | 4 (36.4%) |
| IVc | 12 (37.5%) | 7 (33.3%) | 5 (45.5%) |
| Histological subtype (n = 32) | | | p = 0.5766 |
| Mucosal | 1 (3.1%) | 1 (4.8%) | 0 (0.0%) |
| Cutaneous | 30 (93.8%) | 20 (95.2%) | 10 (90.9%) |
| Other | 1 (3.1%) | 0 (0.0%) | 1 (9.1%) |
| BRAF (n = 31) | | | p = 0.1065 |
| No | 22 (71.0%) | 12 (60.0%) | 10 (90.9%) |
| Yes | 9 (29.0%) | 8 (40.0%) | 1 (9.1%) |
| Missing | 1 | 1 | 0 |
| NRAS (n = 27) | | | p = 0.4475 |
| No | 17 (63.0%) | 9 (56.3%) | 8 (72.7%) |
| Yes | 10 (37.0%) | 7 (43.8%) | 3 (27.3%) |
| Missing | 5 | 5 | 0 |
| Treatment line (n = 32) | | | p = 0.4475 |
| 1 | 23 (71.9%) | 14 (66.7%) | 9 (81.8%) |
| 2 | 6 (18.8%) | 5 (23.8%) | 1 (9.1%) |
| 3 | 3 (9.4%) | 2 (9.5%) | 1 (9.1%) |
| Treatment line (n = 32) | | | p = 0.4414 |
| <2 | 23 (71.9%) | 14 (66.7%) | 9 (81.8%) |
| ≥2 | 9 (28.1%) | 7 (33.3%) | 2 (18.2%) |
| DCI (n = 32) | | | p = 0.4250 |
| Pembrolizumab | 22 (68.8%) | 13 (61.9%) | 9 (81.8%) |
| Nivolumab | 10 (31.3%) | 8 (38.1%) | 2 (18.2%) |

The chi-squared or Fisher's exact test was used to compare categorical variable

dependent reduction of Treg accumulation contributes to slow down Yumm tumor growth and promote total tumor rejection upon ICI therapy.

To investigate the impact of SK1 silencing on CD8 T-cell function, CD8 + T cells were stimulated with PMA and ionomycin in vitro prior to flow cytometric analysis of IFN-γ and TNF production (Supplementary Fig. 6b). We observed that the frequency of IFN-γ CD8 positive cells was increased after SK1 silencing with or without ICI therapy. Of note, the frequency of TNF + CD8 + TILs decreased after SK1 silencing under basal conditions but not upon anti-CTLA-4 or anti-PD-1 treatment. Moreover, the SK1 silencing and anti-PD-1 therapy were associated with an increased expression of CD226 and a reduced expression of TIGIT on CD8 + and CD4 + T cells (Supplementary Fig. 6c, d). Whereas CD226 and TIGIT share the same ligands (CD112 and CD155), they exhibit opposite biological functions: CD226 enhances T-cell activation and TIGIT behaves as a co-inhibitory receptor towards CD8 + TILs[25,26]. We also analyzed the expression of inhibitory ligands, PD-L1 and PD-L2 on tumor and myeloid cells. We found a significant decrease for PD-L1 expression (but not for PD-L2) on tumor cells in shSK1 tumors with and without ICI therapy (Supplementary Fig. 7).

Furthermore, to confirm the beneficial effects of combining SK1 downregulation with ICI in a highly metastatic model with a different genetic background (e.g., BALB/c), we performed experiments with the murine triple-negative breast cancer (TNBC) cells expressing luciferase (4T1-Luc cells). 4T1-Luc cells knocked-down or not for SK1, and exhibiting similar in vitro cell proliferation rates (Fig. 5a, b), were injected in the mammary fat pad and lung metastases were assessed. Whereas lung metastases were significantly reduced with shSK1 4T1 cells as compared with shCtrl 4T1 cells, this phenomenon was further pronounced by combining SK1 silencing and anti-CTLA-4 (Fig. 5c).

In addition, albeit less potently than SK1 silencing, pharmacological inhibition of SK1 by SKI-I[9] significantly improved the anti-CTLA-4 therapy on established Yumm melanoma, and led to a potent increase of tumor rejection and animal survival (Fig. 5d, e), as well as CD8/Treg ratio (Fig. 5f). Under these conditions, SKI-I treatment slightly enhanced the anti-PD-1 efficacy (Fig. 5g). Interestingly, the potentiation of anti-PD-1 efficacy was more pronounced in the MC38 colon cancer model (Fig. 5h).

**SK1 silencing decreases Pges expression in melanoma tumors.** To further characterize the molecular mechanisms triggered in tumor cells upon SK1 inhibition, we used a hypothesis-free, large scale label-free proteomic approach to compare in a global way protein abundance in shCtrl, shSK1(1), and shSK1(2) Yumm cells (Supplementary Fig. 8). Total lysates (three biological replicates for each cell line) were fractionated by one-dimensional (1D) sodium dodecyl sulfate–polyacrylamide gel electrophoresis (SDS-PAGE) and each fraction was analyzed by mass spectrometry on an Orbitrap instrument, leading to the identification and relative quantification of 6197 proteins. Statistical analysis was performed on abundance values measured in samples silenced for SK1 (shSK1(1) + shSK1(2)) vs. shCtrl samples, and 131 proteins were found to exhibit a significant differential expression between the two groups of samples at 1% global FDR. Among these proteins, we found that the prostaglandin E2 synthase (Pges) protein (encoded by *Ptges*) was dramatically downregulated upon SK1 silencing, and appeared as the strongest and most significant variation observed using our unbiased proteomic analysis (Fig. 6a). We further demonstrated by quantitative reverse transcription PCR (RT-qPCR) that this regulation takes place at the mRNA level, as SK1 silencing induced a decreased mRNA expression of *Ptges* (Fig. 6b). Pges is an enzyme

(Fig. 4a), and significantly improved overall survival (Fig. 4b). Of particular interest was the observation that, mice with complete responses fully resisted tumor rechallenge performed almost 2 months after discontinuation of therapy, indicating that downregulation of SK1 in tumors combined with ICI induced efficient long-term memory immune responses and a durable cure of the animals (Fig. 4b). Strikingly, SK1 silencing was significantly associated with a decreased infiltration of Treg and an enhanced CD8/Treg ratio in melanoma tumors and under both control conditions and ICI therapy (Fig. 4c). As compared to ICI alone or SK1 knockdown alone, combined therapies led to a significant additive reduction of Treg content. We observed a strong decrease of CCL17 and CCL22 expression (Fig. 4d). Moreover, Treg depletion using DEREG (Foxp3-DTR-GFP) mice triggered total tumor rejection of either shCtrl or shSK1 Yumm tumors, demonstrating that Treg are a major immunosuppressive population (Supplementary Fig. 5). Altogether, the SK1 silencing-

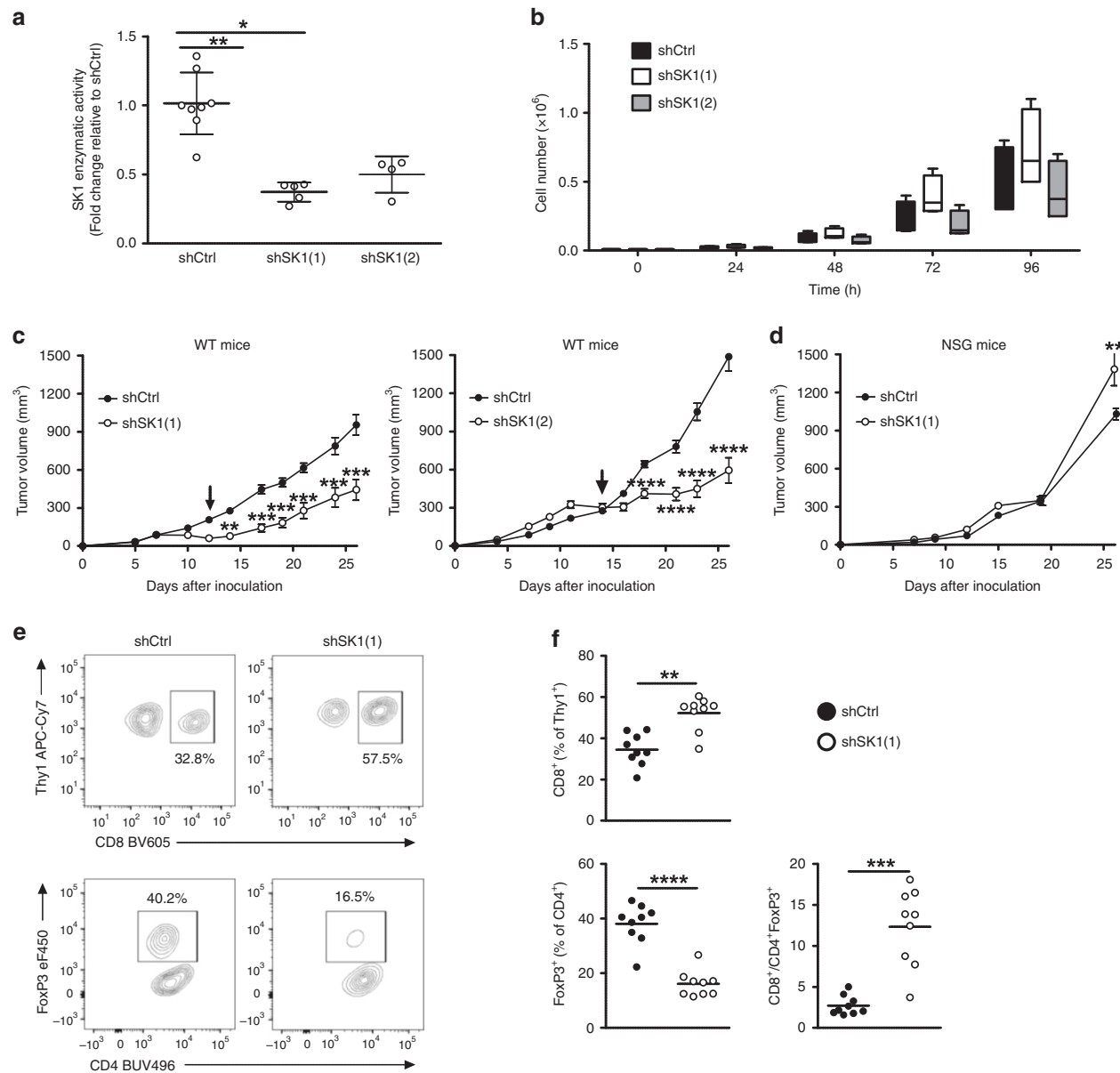

**Fig. 2 SK1 downregulation increases the CD8 + /Treg ratio and reduces tumor growth. a** SK1 enzymatic activity was measured in Yumm cells transfected with a control shRNA (shCtrl, black column) or two SK1-targeted shRNA (white and gray columns) (means ± SEM of 5 independent experiments). **b** Cell proliferation (n = 4). Results represent means of 3 independent experiments. **c** shCtrl (black points), shSK1(1) or shSK1(2) (white points) Yumm cells were injected in wild-type (WT) C57BL/6 mice (n = 12 mice/group). Arrows indicate tumor regression of shSK1 tumors. **d** shCtrl or shSK1(1) Yumm cells were injected in NOD scid gamma (NSG) immunodeficient mice (n = 5 mice/group). Tumor volume was determined at the indicated days. Tumor volumes, presented as means ± SEM, are representative of at least two independent experiments. **e**, **f** shCtrl or shSK1(1) Yumm cells were injected in C57BL/6 mice, and TIL content was analyzed by flow cytometry on day 11. Percentages of CD8 + (upper panel) and CD4 + Foxp3 + (Treg; lower panel) T cells and CD8 + /Treg ratio were calculated. **e** Representative stainings; values indicate the percentage of cells in the quadrant. **f** Each symbol represents an independent tumor (n = 9 mice/group). Results are representative of at least two independent experiments. Data were compared using Kruskal–Wallis test with Dunn's correction (**a**, **b**), two-way ANOVA test (**c**, **d**) or Mann–Whitney test (**e** and **f**).

involved in the biosynthesis of PGE2, a bioactive eicosanoid that plays a complex role in inflammation and cancer[27]. As expected, lipidomic profiling by mass spectrometry showed a strong reduction of both intracellular and extracellular PGE2 levels in Yumm cells after SK1 silencing (Fig. 6c). Of note, SK1 downregulation did not alter the levels of prostaglandin F2α (PGF2α) and prostaglandin D2 (PGD2) (Supplementary Fig. 9), two arachidonic acid metabolites, which, like PGE2, are derived from prostaglandin H2 (PGH2)[28], supporting the specific impact of SK1 silencing on Pges. To exclude an off-target effect of shRNA,

we performed a rescue experiment by transfecting SK1 silencing Yumm cells with a plasmid encoding SK1 (+mSK1). Under these conditions, the inhibitory effect on *Ptges* expression was totally abolished by the rescue of SK1 expression, as well as the in vivo tumor growth (Supplementary Fig. 10).

To investigate the putative role of Pges on ICI efficacy, we generated stable *Ptges* knockdown Yumm cells, using shRNA-mediated technology, which led to a reduction (about 80%) of *Ptges* mRNA expression (Fig. 6d). Of note, downregulation of *Ptges* did not modify *Sphk1* expression, suggesting that SK1 acts

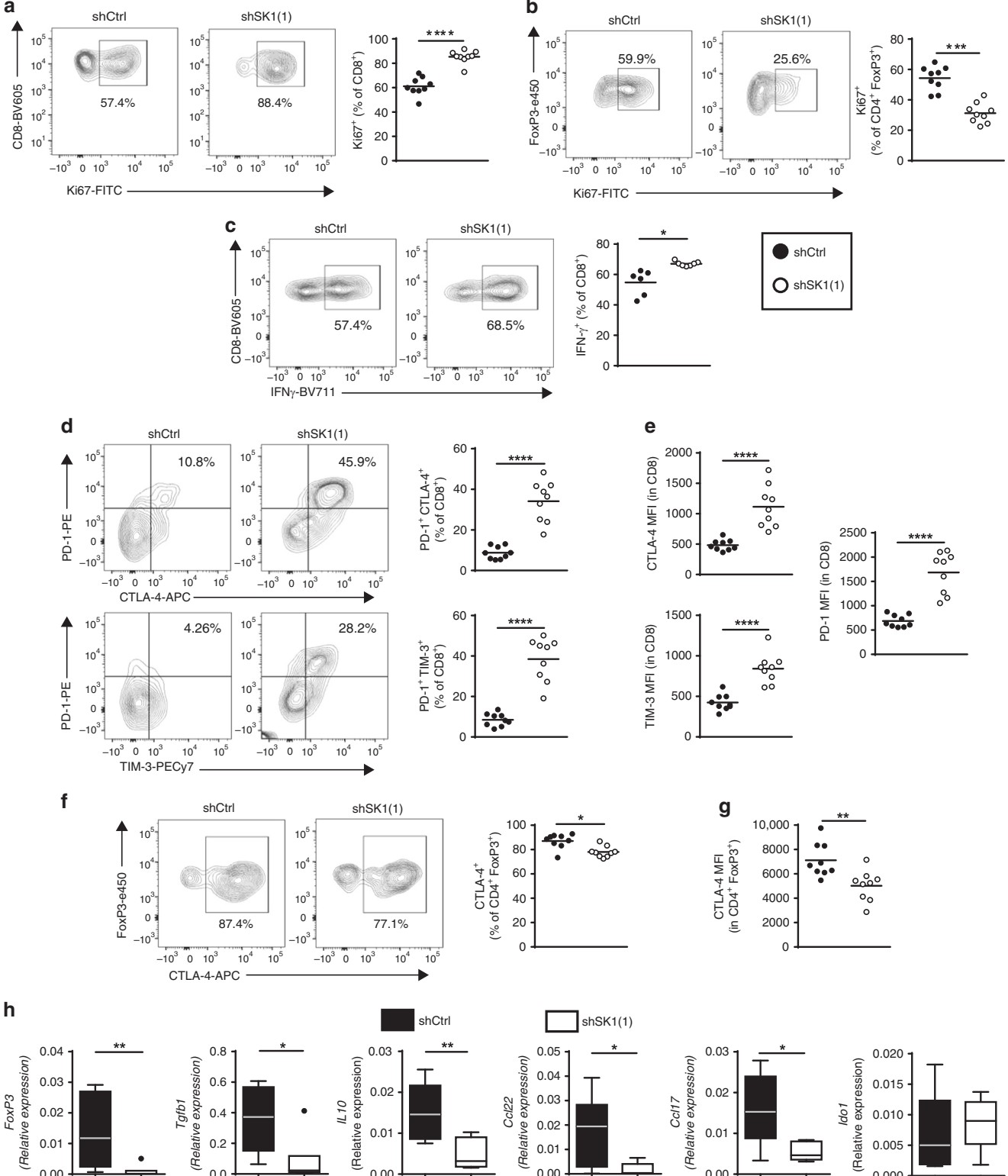

**Fig. 3 SK1 silencing reduces Treg accumulation in tumors.** Control shRNA (shCtrl; black points) or SK1-targeted shRNA (shSK1(1); white points) Yumm melanoma cells were injected intradermally in C57BL/6 mice, and TIL content was analyzed by flow cytometry on day 11 ($n = 9$ mice/group). Percentages (%) of CD8 + T cells (**a**) and Treg cells (**b**) expressing Ki67. **c** Proportion of IFNγ + CD8 cells after PMA and Ionomycin stimulation. **d** Percentage of PD-1 + CTLA-4 + (upper panel) and PD-1 + TIM-3 + (lower panel) cells. **e** MFI of CTLA-4, PD-1 and TIM-3 among CD8 + TILs. Percentage (**f**) and Mean Fluorescence Intensity (MFI; **g**) of CTLA-4 in Treg. Each symbol represents an independent tumor ($n = 9$). **h** Levels of *Foxp3, Tgfb1, Il10, Ccl17, Ccl22* and *Ido1* transcripts in shCtrl (black boxes) or shSK1(1) (white boxes) tumors were quantified by RT-qPCR. The bottom-most and topmost horizontal lines, the lower and upper hinges, and the middle line of the boxplots indicate the minimum and maximum values, the 25th and 75th percentiles, and the median, respectively. Results are representative of two independent experiments ($n = 8$ mice/group). Samples were compared using Mann–Whitney test.

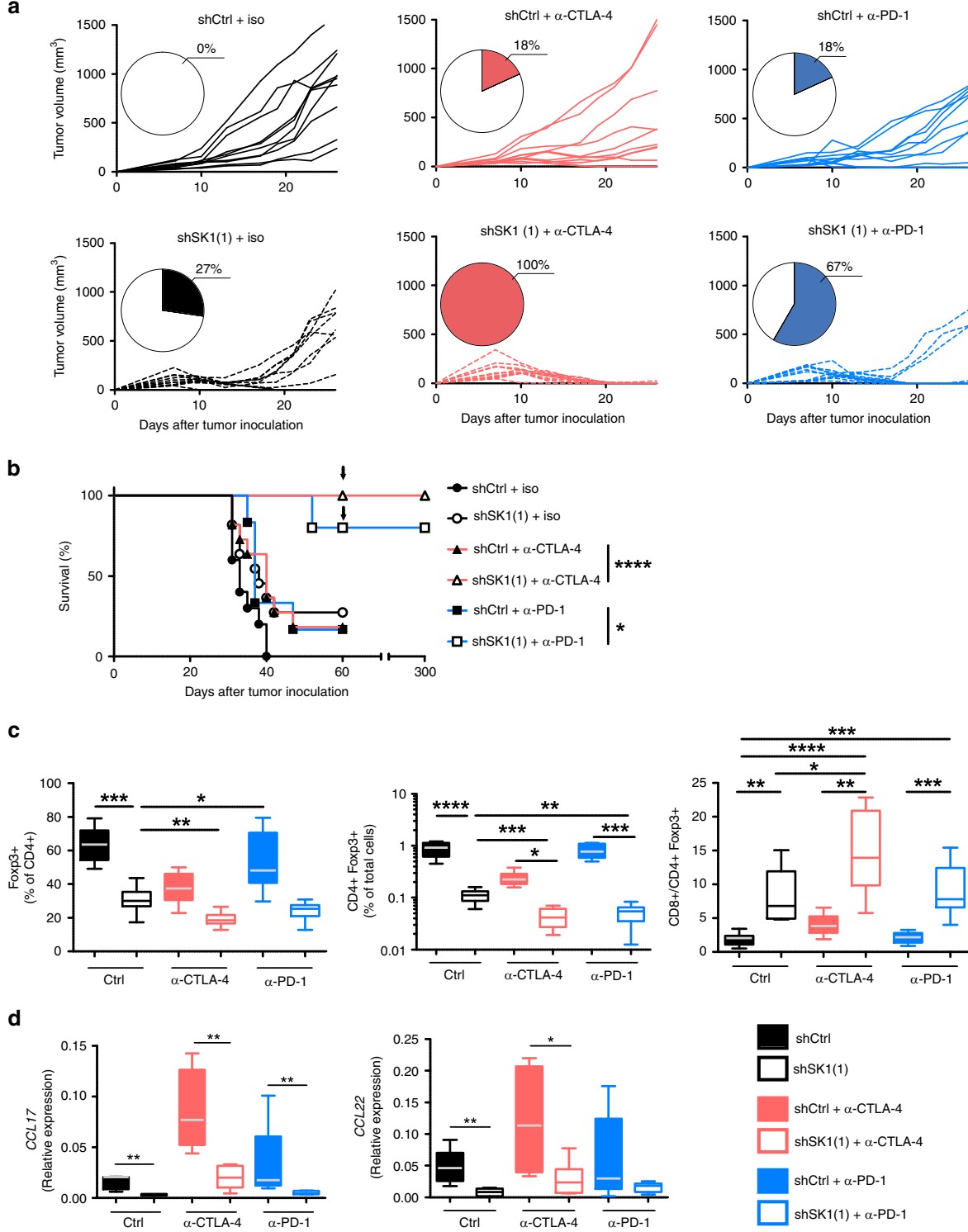

**Fig. 4 SK1 inhibition enhances the efficacy of anti-CTLA-4 or anti-PD-1 therapy.** Control shRNA (shCtrl) or SK1-targeted shRNA (shSK1(1)) Yumm cells were injected on day 0, and then mice were treated with isotype (iso) control antibody (black lines; shCtrl $n = 10$ mice, shSK1(1) $n = 11$ mice), anti-CTLA-4 (red lines; shCtrl $n = 11$ mice, shSK1(1) $n = 11$ mice), or anti-PD-1 (blue lines; shCtrl $n = 11$ mice, shSK1(1) $n = 12$ mice). **a** Individual growth curves are depicted for each tumor. Inserts: numbers indicate percentage (%) of tumor-free mice at day 26. **b** Kaplan–Meier survival curves with log-rank test (iso, circles and black lines; anti-CTLA-4, triangles and red lines; anti-PD-1, squares and blue lines. Full and empty symbols correspond to shCtrl and shSK1(1), respectively). The arrow indicates a second orthotopic injection of shSK1 cells. **c** Percentages of CD4 + Foxp3 + among CD4 T cells (left panel) or total cells (middle panel) and CD8/CD4 + Foxp3 + T-cell ratio (right panel) at day 11 (iso, black/white boxes; anti-CTLA-4, red boxes; anti-PD-1, blue boxes. Full and empty symbols correspond to shCtrl and shSK1(1), respectively) ($n = 9$ mice/group). **d** *CCL17* and *CCL22* mRNA expression in tumors. Results are representative of two independent experiments ($n = 5$ mice/group). **c**, **d** Samples were compared using Kruskal–Wallis test with Dunn's correction.

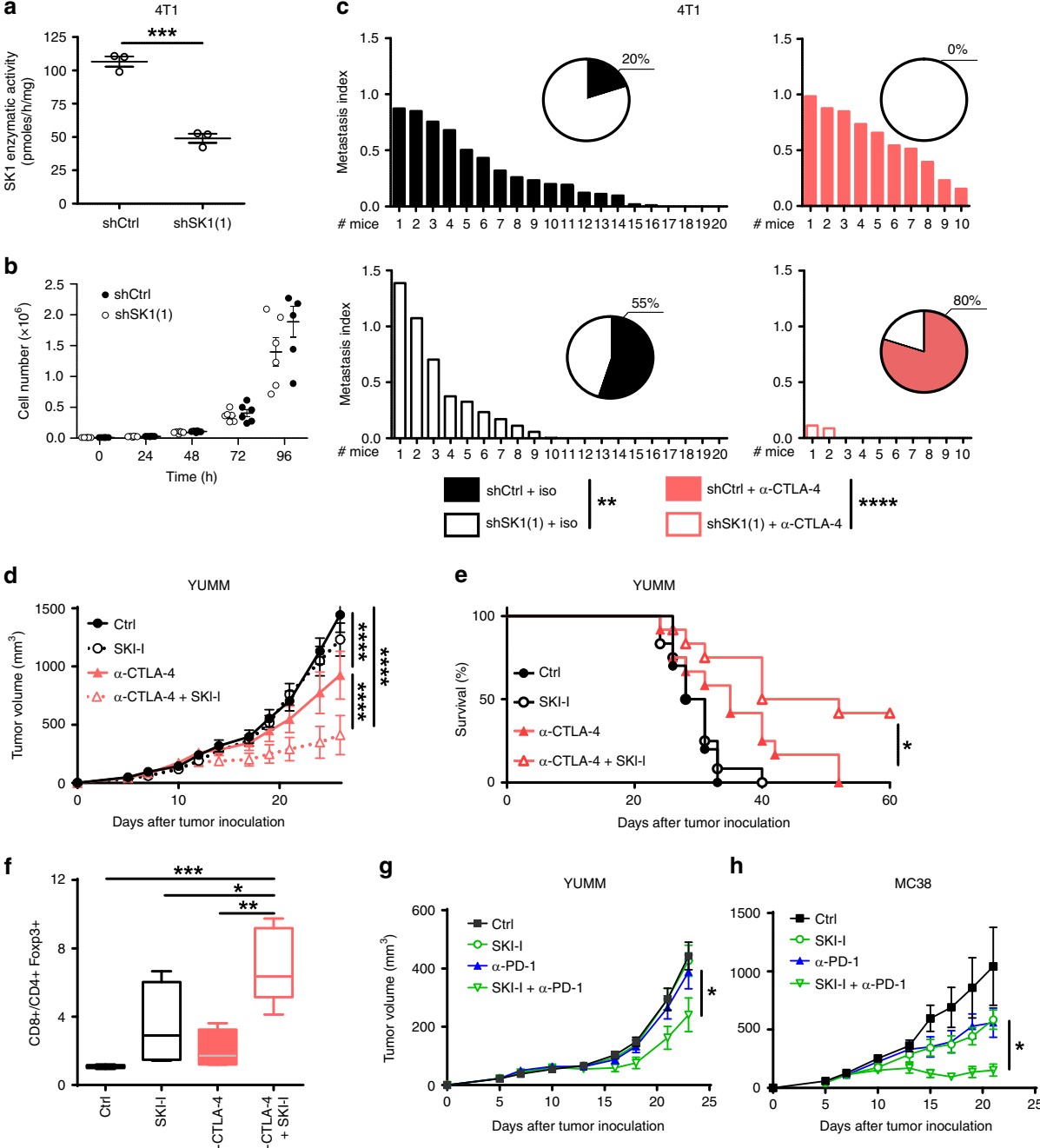

**Fig. 5 SK1 targeting enhances ICI efficacy in various cancer models. a** SK1 enzymatic activity in 4T1-Luc cells stably transfected with a control shRNA (shCtrl;) or a SK1-targeted shRNA (shSK1(1)). Data are means ± SEM of three independent experiments. **b** Cell proliferation. Results are means ± SEM of four independent experiments. **c** shCtrl and shSK1(1) 4T1-Luc cells were injected in mammary fat pad of BALB/C mice. Mice were then treated with isotype control antibody (iso) or α-CTLA-4. Metastasis index at day 31 in lung of mice injected with shCtrl (upper panel; full columns) or shSK1(1) (lower panel; empty columns) 4T1-Luc tumors and treated (right panel; red columns) or not (left panel; black and white columns) with α-CTLA-4. Inserts: numbers indicate percentage of metastasis-free mice. Data are representative of two independent experiments. **d**–**f** Mice were challenged with untransfected Yumm cells and then treated or not with vehicle (Ctrl; black circles with black full line, $n = 11$ mice), SKI-I (white circles with black dotted line, $n = 12$ mice), anti-CTLA-4 (red and white triangles; full symbols with full line (without SKI-I $n = 12$ mice) or empty symbols with dotted line (with SKI-I $n = 12$ mice). Data are representative of two independent experiments. **d** Tumor volumes are presented as means ± SEM. **e** Kaplan–Meier survival curves with log-rank test. **f** CD8/CD4 + Foxp3 + T-cell ratio in tumors at day 11 ($n = 4$). **g** Mice were challenged with untransfected Yumm cells and then treated or not with vehicle (Ctrl; black squares with black line), SKI-I (empty green circles or triangles with green lines) and anti-PD-1 (full blue (without SKI-I) or empty green (with SKI-I) triangles) ($n = 12$ mice/group). **h** Mice were inoculated with MC38 and then treated or not with vehicle (Ctrl; black squares with black line), SKI-I (empty green circles or triangles with green lines) and anti-PD-1 (full blue (without SKI-I) or empty green (with SKI-I) triangles) ($n = 5$ mice /group). Data are representative of two independent experiments. Tumor volumes are presented as means ± SEM. Samples were compared using Mann–Whitney test (**a**) Kruskal–Wallis test with Dunn's correction (**b**, **c**, and **f**) or using two ANOVA test (**d**, **g**, and **h**).

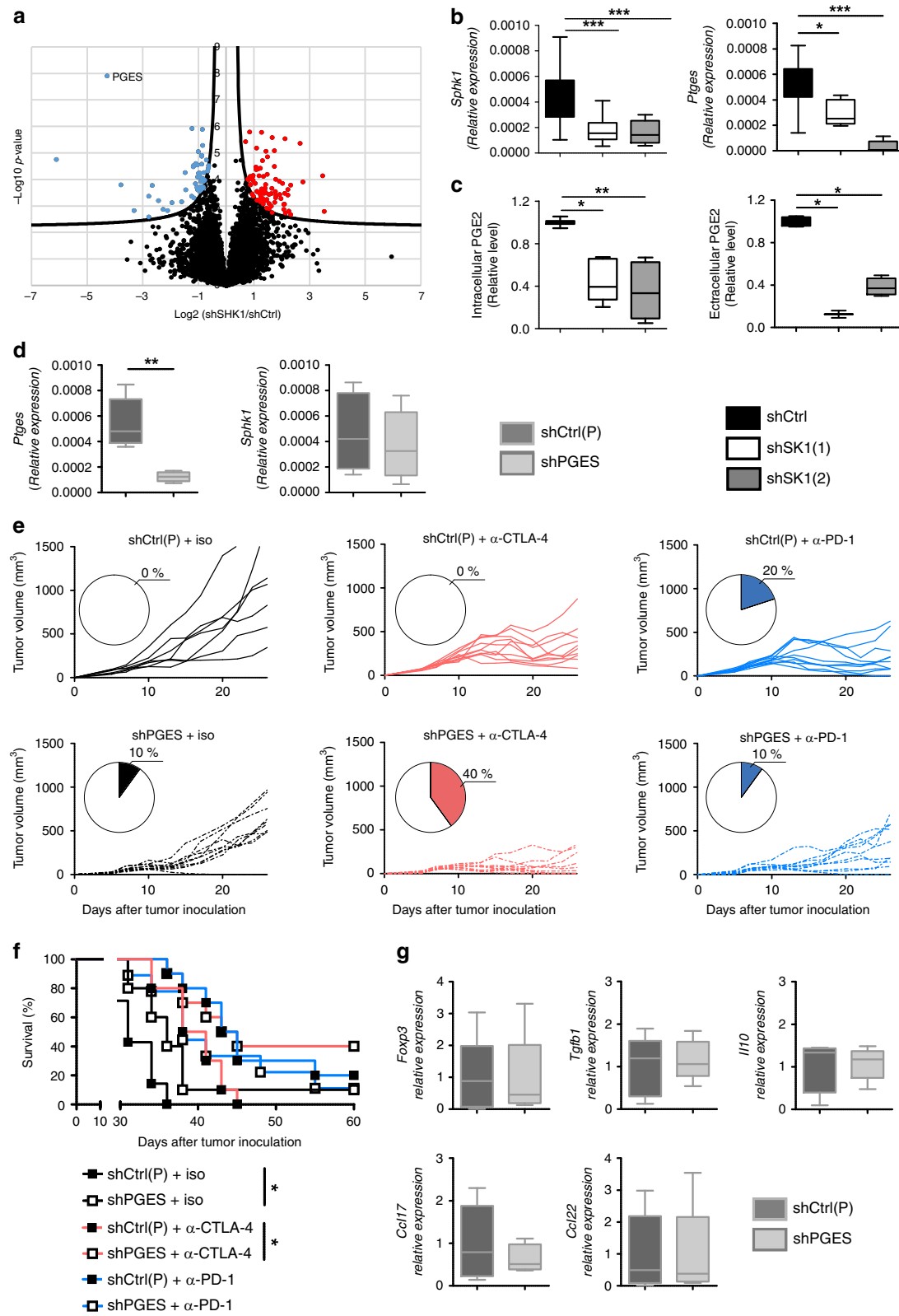

upstream of Pges in the signaling cascade. Then, Yumm cells, in which Pges was silenced (shPges) or not (shCtrlP), were intradermally injected in C57BL/6 mice. Our data show that Pges silencing slightly potentiated CTLA-4 blockade and promoted tumor rejection in 40% of animals. In sharp contrast, Pges silencing failed to improve anti-PD-1 therapy (Fig. 6e, f).

These results indicate that Pges expression may contribute, at least in part, to overcome the resistance of melanoma to anti-CTLA-4 but not anti-PD-1 therapy. Moreover, Pges silencing did not reduce the expression of Foxp3, IL-10, TGF-β, CCL17 and CCL22 (Fig. 6g). Altogether, our data demonstrate non-fully redundant immunosuppressive functions of SK1 and Pges, and

**Fig. 6 Reduced PGE2 production induced by shSK1 partially enhances ICI efficacy. a** Volcano plot showing proteins differentially regulated between control shRNA (shCtrl) and SK1-targeted shRNA (shSK1(1 and 2)) Yumm cells. Blue and red points represent under- and overexpressed proteins, respectively. **b** *Sphk1* (left panel) aexpression in shCtrl (black column, $n = 17$ mice), shSK1(1) (white column, $n = 17$ mice) or shSK1(2) (gray column $n = 12$ mice) Yumm cells and *Ptges* (right panel) expression in shCtrl (black column, $n = 8$ mice), shSK1(1) (white column, $n = 8$ mice) or shSK1(2) (gray column, $n = 5$ mice) Yumm cells; $n = 5$–10 independent experiments. **c** Intracellular (left panel, $n = 5$) and extracellular (right panel, $n = 4$) PGE2 levels were normalized to those in shCtrl; $n = 2$–6 independent experiments. **d** *Ptges* (left panel, $n = 5$) and *Sphk1* (right panel, $n = 7$) expression was analyzed in shCtrl (P) (dark gray columns) or shPGES (clear gray columns) Yumm cells; $n = 5$–6 independent experiments. **e, f** Mice were challenged with shCtrl(P) or shPGES cells on day 0, and then treated with isotype control antibody (iso; black lines), anti-CTLA-4 (red lines) or anti-PD-1 (blue lines). **e** Individual growth curves are depicted for each tumor ($n = 10$ mice). Insets: numbers indicate percentage of tumor-free mice at day 25. **f** Kaplan–Meier survival curves with log-rank test. shCtrl(P) and shPGES tumors were represented by black and white squares, respectively. **g** Levels of *Foxp3, Tgfb1, Il10, Ccl17* and *Ccl22* transcripts in shCtrl(P) (dark gray columns) or shPGES (clear gray columns) tumors ($n = 5$). Data, expressed as relative expression, are means ± SEM. The bottom-most and topmost horizontal lines, the lower and upper hinges, and the middle line of the boxplots indicate the minimum and maximum values, the 25th and 75th percentiles, and the median, respectively. Samples were compared using Kruskal–Wallis test with Dunn's correction (**b** and **c**) or Mann–Whitney test (**d**).

that SK1 silencing leads to a better therapeutic benefit in combination with immunotherapy compared to Pges silencing in melanoma.

**SPHK1 expression in melanomas correlates with immune escape**. To analyze the translational relevance of our findings in patients affected with metastatic melanoma, we correlated *SPHK1* transcript levels with those of transcripts encoding immunosuppressive factors, using a publically available dataset (TCGA) of human melanoma biopsies[29]. Interestingly, samples with high *SPHK1* expression showed a higher expression of immune escape genes, including *PTGES, FOXP3, TGFB1, IL10, CCL22, CCR4, IDO1, IDO2, CTLA4, PDCD1* (encoding PD-1), *CD274* (encoding PD-L1), *PDCD1LG2* (encoding PD-L2), *TIGIT, LAG3* and *HAVCR2* (encoding TIM-3) (Fig. 7a, b). Altogether, these data indicate that *SPHK1* overexpression is associated with an immunosuppressive TME in humans, in line with our preclinical findings.

## Discussion

This study provides the first evidence that melanoma SK1 behaves as an immune escape lipid kinase, leading to an increased expression of immunosuppressive factors in the TME and impairing the response to anti-CTLA-4 or anti-PD-1 treatment.

SK1 overexpression has been described in many different cancer types, including lung[30], gastric[31–33], breast cancer[34,35], and glioblastoma[36]. High SK1 expression correlates with decreased survival of patients with various cancers in a meta-analysis of clinical studies[37]. However, its role in melanoma patient survival has not yet been described. Herein, we show that high SK1 expression in melanoma cells is associated with anti-PD-1 resistance in patients with advanced melanoma. Considering the small number of patients, the association between the expression of SK1 and the tumor stage, the mutation status of BRAF and NRAS, as well as immune responses could not be performed due to weak statistical power in this retrospective study. This will be performed on a larger cohort of advanced melanoma patients treated with anti-PD-1 in combination or not with anti-CTLA-4, from a prospective clinical trial (IMMUSPHINX: NCT03627026) we are currently conducting in our institute.

SK1 produces the phospholipid S1P, which exerts its functions through intracellular actions or ligation to five cell surface G-protein-coupled receptors (named S1PR1-5) expressed both on cancer cells and cells of the TME[38,39]. The SK1/S1P/S1PR axis can regulate tumor cell behavior, as well as the TME composition[8–11]. For instance, it promotes tumor angiogenesis[40]. Thereby, neutralization of extracellular S1P with an anti-S1P antibody significantly inhibited bFGF- and VEGF-induced angiogenesis, as

well as growth of different tumors including melanomas[41]. The present study demonstrates that downregulation of SK1 reduced melanoma growth in wild-type but not immunodeficient mice or in CD8 cell-depleted mice, highlighting the involvement of CD8 + T cells in this phenomenon. Although a potential role for SK1 silencing on tumor angiogenesis inhibition cannot be excluded, it is unlikely the main process delaying tumor growth in our model.

To the best of our knowledge, this study provides the first evidence that tumor SK1 plays a key role in the modulation of TIL composition, leading to potent Treg accumulation. Indeed, we show that SK1 targeting combined with ICI resulted in a high CD8/Treg ratio. These findings highlight the major role played by tumor SK1 in immune escape. Thus, inhibiting SK1 may represent an original strategy to enhance ICI response. As targeting SK1 specifically in melanoma cells is not currently achievable, it became important to document the putative benefit of systemic pharmacological approaches based on the combination of a SK1 inhibitor and ICI. As documented here, SK1 inhibition significantly sensitized melanoma to ICI. Although, a single melanoma cell line was tested here, two other cancer models, i.e, colon and breast cancers, were shown to respond to the combination of ICI and SK1 targeting.

In line with our findings, previous studies demonstrated that the S1P/S1PR1 axis modulates the anti-tumor immune response. First, S1PR1 signaling in T cells enhances the tumor infiltration of Treg in a STAT3-dependent manner, reducing CD8 + TIL and increasing breast cancer and melanoma growth in mice[42]. Second, the role of S1P signaling in metastatic colonization and immune response was also illustrated through SPNS2 targeting[43]. SPNS2 is a S1P transporter that regulates the S1P levels in blood and lymph[44]. The deletion of SPNS2 either globally or in a lymphatic endothelial-specific manner decreased the number of pulmonary metastatic melanoma foci[43], a phenomenon associated with an increased infiltration of activated CD8 + T cells and NK cells in the lung. Of note, the role of S1P signaling on ICI efficacy has not yet been addressed.

Further analyses show that SK1 knockdown in melanoma tumors potently reduced the production of various immunosuppressive cytokines such as TGFβ, IL10, CCL17, and CCL22, which is in line with the strong decrease of Treg tumor infiltration. To get insights into the molecular mechanisms underlying, we performed a proteomic analysis of SK1 knockdown and control melanoma cells. Pges was found to be dramatically decreased upon SK1 silencing, leading to a strong reduction of PGE2 production. It was previously demonstrated that SK1 downregulation attenuates PGE2 production through inhibition of PGES, but not COX2, via the p38 MAPK pathway in human umbilical vein endothelial cells[45]. Whereas in our mouse

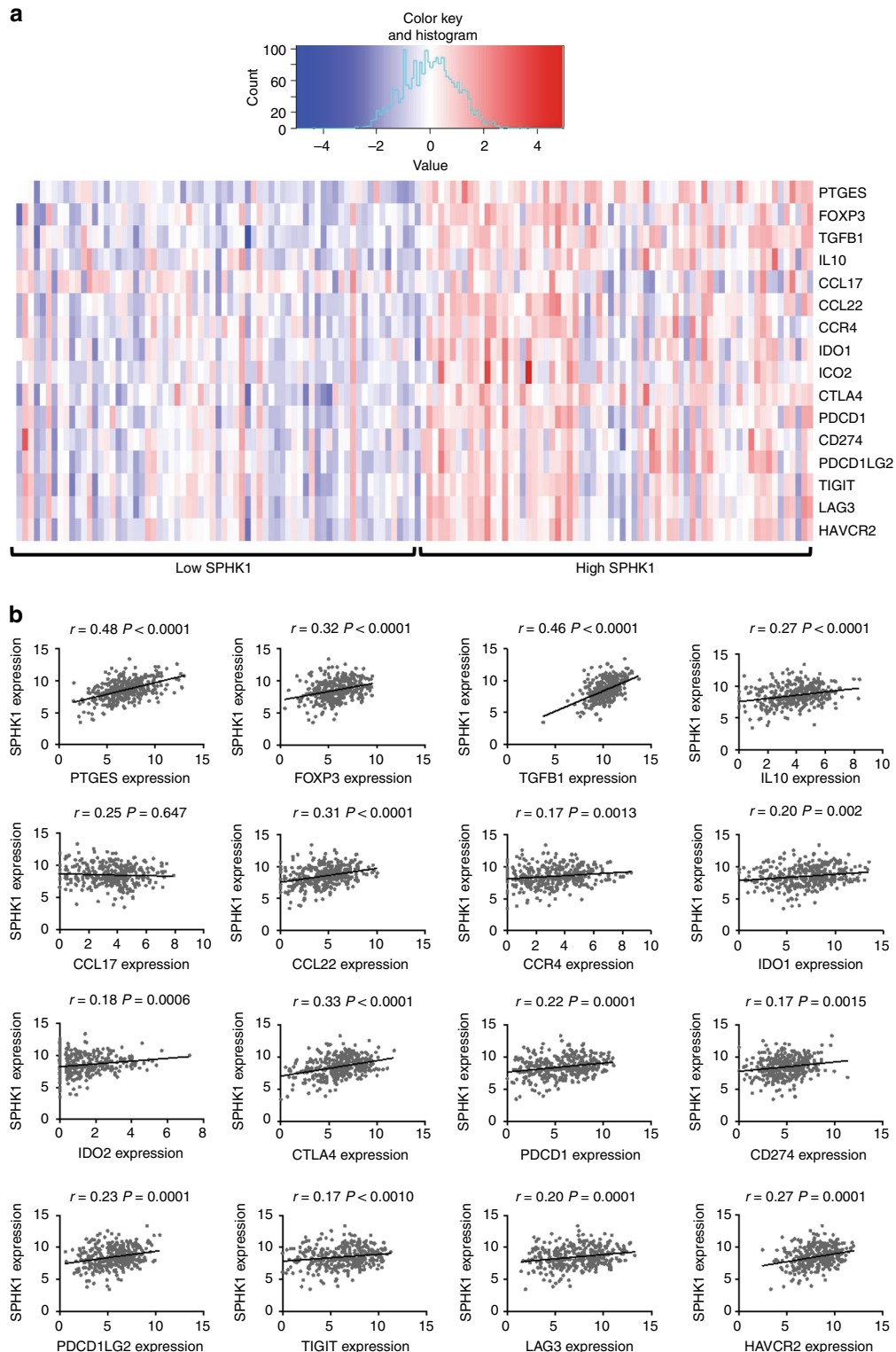

**Fig. 7 Human *SPHK1* expression correlates with immune suppressive factors. a** Heat map for a selected list of genes indicative of immune suppression exhibiting low (20th percentile) and high (80th percentile) *SPHK1* expression in melanoma samples using the TCGA cohort. Red indicates high expression and blue indicates low expression relative to the mean expression of the gene across all samples ($n = 136$). **b** Correlations between *SPHK1* expression (log2($x + 1$) transformed RSEM normalized count) and genes indicative of immune suppression using non-parametric Spearman's test ($n = 342$).

melanoma model Pges knockdown sensitized, albeit to a lesser extent than SK1 knockdown, to anti-CTLA-4 therapy, no effect towards anti-PD-1 treatment was observed. This suggests that inhibition of Pges expression, along SK1 knockdown in melanoma cells, is probably not the sole molecular mechanism

responsible for the synergy observed between SK1 silencing and ICI. Our data seem at odds with a recent study, which shows that the inhibition of PGE2 production, by aspirin or celecoxib (two COX inhibitors), increased anti-PD-1 treatment efficacy in mouse melanoma and colon cancer[46]. One should note however,

that the pharmacological inhibition of COX-driven synthesis of PGE2 is likely associated with the decreased production of PGH2 and additional eicosanoid metabolites such as $PGI_2$, $PGD_2$, $PGF_{2a}$, and $TXA_2$, which may also influence platelet aggregation, as well as immune responses[47–49].

In conclusion, our current observations demonstrate that targeting SK1 is a promising option as inhibition of this specific lipid kinase affects (i) the primary tumor growth, (ii) the intratumoral CD8/Treg ratio (iii) the immunotherapy efficacy by immune checkpoint blockade such as anti-CTLA-4 or anti-PD-1. Future work is needed to explore the potential synergism of SK1 antagonism with other ICI or with cancer vaccine. Moreover, the potential roles of other immune cell subsets like NK, γδ T cells and especially various dendritic cells, as well as the role of SK1 in a non-T-cell-inflamed tumors have not yet been explored. These will require further investigation.

Taken together, our findings support the notion that SK1 is a checkpoint lipid kinase, which modulates immune escape mechanisms. Also, the relative abundance of SK1 within tumors could be highly predictive of the response to ICI therapy. Finally, it is tempting to speculate that clinical strategies based on combining SK1 inhibition and ICI may improve the response rate in patients affected with melanoma or other malignancies.

## Methods

**In silico analysis with Oncomine database.** *SPHK1* expression in human nevi or melanomas was analyzed using the Oncomine tool (www.oncomine.org) in 2 different cohorts[50,51].

**In situ mRNA hybridization.** In situ detection of *SPHK1* transcripts in formalin-fixed, paraffin-embedded tissues was performed using the RNAscope assay with RNAScope 2.5 vs. Probe-Hs-SPHK1 and the ACD RNAscope 2.0 Red kit (Advanced Cell Diagnostics). Assay specificity was assessed measuring the signal in positive and negative control samples. Positivity of endothelial cells was used as an intrinsic positive control. Cases with positive intrinsic control and no signal in tumor cells were considered as negative. We determined the percentage of positive tumor cells blinded to clinical response to treatment.

**Murine cell lines.** Yumm 1.7 melanoma cells derived from Braf$^{V600E/wt}$ Cdkn2A$^{-/-}$ Pten$^{-/-}$ mice were a gift from Dr. M. Bosenberg (Yale University School of Medicine, New Haven, CT) and were described elsewhere[18,19]. Yumm cell lines were cultured in Opti-MEM supplemented with 10% FCS. To guarantee cell line authenticity, Yumm cell lines were used for a limited number of passages and routinely tested for the expression of melanocyte-lineage proteins such as MelanA/MART1. 4T1-luc cell line was kindly provided by Pr. H. Prats (CRCT, INSERM U1037, Toulouse, France). MC38 cell line was kindly provided by Drs. T. Chardès and A. Pèlegrin (INSERM U1194, Montpellier, France). 4T1 and MC38 were cultured in FCS Dulbecco's modified Eagle medium supplemented with 10% FCS and L-glutamine. Cell lines were routinely tested for the absence of mycoplasma contamination by PCR.

**Cell transfection.** Yumm cells were co-transfected, in a 1:10 ratio, with the pEGFP-N empty vector and a SK1 shRNA (shSK1(1): GCACCCAAAC-TACCTTTGGAT or shSK1(2): GCAGGTGACTAATGAAGACCT) plasmid (shRNA from Thermoscientific) or a control non-targeting shRNA (shCtrl) plasmid (pLK01, Addgene). In other experiments, Yumm cells were transfected with a Ptges shRNA (shPtges) construct in a retroviral GFP vector or non-effective 29-mer scrambled shRNA (shCtrl(P); Origene). Cells were transfected using Lipofectamine 2000 reagent (Invitrogen). Transfected shSK1 and shCtrl cells were selected with puromycin and G418. shPtges and shCtrl(P) cells were selected only with puromycin. Finally, GFP-expressing cells were sorted by FACS.

**SK1 enzymatic assay.** SK1 activity was determined as described[52] with minor modifications.

**Yumm tumor challenge and treatments.** Animal experiments were conducted in accordance with national and international policies, and our protocol was approved by the Regional Ethics Committee of Midi-Pyrénées. Untransfected or transfected Yumm cells (3.10$^5$) were intradermally injected into the flank of 7-week-old C57BL/6 mice (Charles River, L'Arbresle, France) or NSG mice (a kind gift from Dr. JE. Sarry, INSERM U1037, Toulouse, France). Tumor volumes were measured every 2-3 days using a caliper[8]. For ICI combination experiments, mice were injected i.p. three times with anti-CTLA-4 (200 μg per mouse on day 7 and 100 μg per mouse on day 10 and 13), or with anti-PD-1 or isotype control antibody

(200 μg per mouse on day 5, 7 and 10). Anti-CTLA-4 (clone 9H10), anti-PD-1 (clone RMP1-14) and isotype control antibodies (clone 2A3) were purchased from BioXcell. For SKI-I treatment, mice were treated or not with 50 mg/kg SKI-I (N'-[(2-hydroxynaphthalen-1-yl) methylidene]-3-(naphthalen-2-yl)-1H-pyrazole-5-carbohydrazide, Enamine) by i.p. injection on days 5, 7, 10, 13, and 15.

**4T1 Metastasis analysis.** At day 31, mice with 4T1-Luc shCtrl or shSK1 tumors and treated or not with anti-CTLA-4 were sacrificed. lungs were collected and homogenized using Precellys. RT-qPCR was performed using primers for transcripts encoding murine Luciferase and Cyclophilin-A (Qiagen). Evaluation of the metastatic load was performed by quantification of luciferase mRNAs in lungs by RT-qPCR and normalized to Cyclophilin-A. Luciferase_F CTCACTGAGACTAC ATCAGC; Luciferase_R TCCAGATCCACAACCTTCGC.

**MC38 tumor growth.** Mice were inoculated with 300,000 MC38 cells at day 0, and then treated or not with vehicle (Ctrl), SKI-I (50 mg/kg by i.p. injection on days 5, 7, 10, 13, and 15) and anti-PD-1 (100 μg per mouse at day 7 and 10).

**Antibodies and flow cytometry.** Yumm cells (3.10$^5$) were intradermally injected into C57BL/6 mice. At day 11, mice were sacrificed and tumors were collected and digested with Mouse Tumor Dissociation kit and GentleMacs (Miltenyi). Cells were counted and directly stained with LIVE/DEAD reactive dyes (Invitrogen, 1:100) and the indicated antibodies: anti-mouse CD45 (clone 30-F11, 1:200), anti-mouse CD8 (clone 53-6.7, 1:200), anti-mouse CD4 (clone GK1.5, 1/200), anti-mouse Ki67 (clone B56, 1:5) and anti-mouse IFN-γ (clone XMG1.2, 1:100) are from BD Biosciences; anti-mouse Foxp3 (clone FJK-16s, 1:50), anti-mouse CTLA-4 (clone UC10-4B9, 1:200) and anti-mouse PD-1 (clone J43, 1:200) are from Thermo Fisher; anti-mouse TIM-3 (clone 1G9, 1:200), anti-mouse TIGIT (clone 1G9, 1:200), anti-mouse CD226 (clone TX42.1, 1:200) and anti-mouse Thy1 (clone 30-H12, 1:400) are from Biolegend. Cell suspensions were incubated with anti-CD16/CD32 blocking antibodies (Thermo Fisher) prior to incubation. For IFN-γ analysis, tumor cells were incubated with a Cell Stimulation Cocktail (Invitrogen). Cells were analyzed with a BD LSR Fortessa X-20 (BD Biosciences), followed by Flow-Jo software analysis. Gating strategies followed for flow cytometry analysis are shown in Supplementary Fig. 1.

**RT-qPCR analysis.** At day 11, mice were sacrificed and tumors were collected and dissociated using a Precellys Evolution tissue homogenizer (Bertin technologies). RNA was extracted using the RNeasy Midi Kit (Qiagen). Complementary DNA was synthetized with SuperScript II Reverse Transcriptase (Thermo fisher). qPCR was performed using SYBR Green Master Mix (Takara). Quantification of mRNAs was normalized to the expression of cyclophilin-A (cyclo-A) or β-actin as reference genes. Prevalidated primers for transcripts encoding murine, beta-actin (QT00095242), CCL22 (QT00108031), CCL17 (QT00131572), SK1 (QT01046395), Ptges (QT00118223), IDO1 (QT00103936) were purchased form Qiagen. The primers for IL-10 and TGF-β1 were as follows: TGFβ1_F 5′AGCTGCGGCTTGCA GAGATTA3′, TGFβ1_R 5′TGCCGTACAACTCCAGTGAC3′, IL10_F 5′CGGG AAGACAATAACTGCACC3′, IL10_R 5′TTTCCGATAAGGCTTGGCAAC3′ CycloA_F 5′GTCAACCCCACCGTGTTCTT3′, CycloA_R 5′CTGCTGTCTTTGG GACCTTGT3′

**Proteomic analysis.** Whole-cell extracts of Yumm cells transfected either with shCtrl, shSK1(1) or shSK1(2) (three replicates each) were prepared by direct lysis in Tris 50 mM, SDS 2% and sonication. Proteins were reduced (DTT 5 mM, 95°C, 5 min), alkylated with iodoacetamide 100 mM, fractionated into ten bands on a 1D-SDS gel, and in-gel digested with trypsin. Peptides were analyzed by liquid nanochromatography (UltiMate 3000 RSLCnano system, Dionex, Amsterdam, The Netherlands) coupled to an LTQ-Velos mass spectrometer (ThermoScientific, Bremen, Germany) as described[53]. Raw MS files were processed using MaxQuant (v 1.5.3.30) for database search against Mouse entries of the Swissprot database, and validation through the target-decoy approach at a false-discovery rate of 5% and 1% for peptides and proteins, respectively. Label-free relative quantification of the proteins across samples was performed in MaxQuant using the LFQ algorithm, and the Perseus software was used for missing value imputation and statistical analysis. A two-sided Student-t-test with variance correction (S0 = 0.1) was calculated between the shCtrl group (three replicates) and SK1-silenced group (shSK1 (1) and shSK1(2), six replicates), and the global false-discovery rate was adjusted at 1% using the Perseus permutation–based method to identify proteins significantly regulated between the two groups.

**Quantification of eicosanoids.** Eicosanoid concentrations in conditioned media or Yumm cell lysates were determined by ultra-performance liquid chromatography, using an Agilent 1290 system coupled to a G6460 triple quadripole mass spectrometer (Agilent Technologies).

**Analysis of *SPHK1* expression in human melanoma.** *SPHK1* expression was analyzed using the TCGA melanoma cohort. TCGA genomic and clinical data were downloaded from the UCSC Xena platform (https://xenabrowser.net/). The

analysis population consisted of 342 patients with distant metastasis for whom RNAseq and clinical data overlap. No formal sample size calculation was performed concerning TCGA analysis. Gene expression was measured experimentally using the Illumina HiSeq 2000 RNA Sequencing platform and $\log_2(x+1)$ transformed. The strength of the relationship between *SPHK1* transcript levels with those of transcripts encoding immunosuppressive factors was assessed using Spearman's rank correlation coefficient.

**Statistical analyses**. Results are expressed as means ± SEM, and group comparisons were performed with Mann-Withney, Kruskal–Wallis test with Dunn's correction or two-way Anova repeated measures with Bonferoni correction for multiple comparisons. The strength of relationship between continuous biomarkers was assessed using Spearman's rank correlation coefficient. The major clinicopathological characteristics and available treatment information of the cohorts are presented in Table 1 and summarized by frequency and percentage. All survival times were calculated from the initiation of immunotherapy and estimated by the Kaplan–Meier method with 95% confidence intervals (CI), using the following first event definitions: progression or death from any cause for progression-free survival (PFS) and death from any cause for overall survival (OS). Patients still alive were censored at their date of last follow-up. Comparison between groups (low expression vs. high expression) was performed using log-rank test. All tests were two-sided and differences were considered statistically significant when $p < 0.05$ (*$p < 0.05$; **$p < 0.01$; ***$p < 0.001$, ****$p < 0.0001$). Data were analyzed using GraphPad Prism (GraphPad Software, San Diego, CA) and STATA v13 (Stata Corporation, College Station, TX, USA).

**Reporting summary**. Further information on research design is available in the Nature Research Reporting Summary linked to this article.

## Data availability

The mass spectrometry proteomics data have been deposited to the ProteomeXchange Consortium via the PRIDE partner repository with the dataset identifier PXD016535 [http://www.ebi.ac.uk/pride/archive/projects/PXD016535]. The oncomine data referenced during the study in Fig. 1 are available in a public repository from the oncomine website (www.oncomine.org). Talantov data can be accessed here. Xu data can be accessed here. The TCGA source data underlying Fig. 7 are available in UCSC Xena platform: https://tcga.xenahubs.net/download/TCGA.SKCM.sampleMap/HiSeqV2.gz. All the other data supporting the findings of this study are available within the article and its supplementary information files and from the corresponding author upon reasonable request. A reporting summary for this article is available as a Supplementary Information file.

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

## Acknowledgements
This work was supported by INSERM, Paul Sabatier University, Ligue Nationale Contre le Cancer (LNCC, Equipe Labellisée 2013), INSERM Transfert, Cancéropôle Grand Sud-Ouest, ROTARY Toulouse clubs, Fondation Toulouse Cancer Santé and Fondation ARC (IMMUSPHINX project). The research leading to these results has received funding from the Transcan-2 Research Program, which is a transnational R&D program jointly funded by national funding organizations within the framework of the ERA-NET Transcan-2. C.I. is a recipient of a fellowship from LNCC and Fondation pour la Recherche Médicale (FRM). The IBiSA Toulouse Proteomics facility is supported by the following institutions: Région Midi-Pyrénées, Fonds Européens de Développement Régional, Toulouse Métropole, and the French Ministry of Research with the 'Investissement d'Avenir Infrastructures Nationales en Biologie et Santé program' (Proteomics French Infrastructure project, ANR-10-INBS-08). P.B. is supported by the LABEX TOUCAN. The authors would like to thank Drs. J. Riond and L. Martinet for critical reading (CRCT, Inserm UMR1037, Toulouse, France) and Drs. N. Cénac (IRSD, Inserm U1220, Toulouse, France), JJ. Fournié, and M. Ayyoub (CRCT, Inserm UMR1037, Toulouse, France) for fruitful discussion. We are grateful to the animal (CREFRE, Toulouse, France) and flow cytometry (Inserm UMR1037, Toulouse, France) facilities. We also thank Dr. JE Sarry (CRCT, Inserm UMR1037, Toulouse, France) for the kind gift of NSG mice and Dr. C. Lachaud (CRCM; Marseille, France) for the generation of stable cell line using the CRIPSR/Cas9 technology. We acknowledge the assistance of Drs. P. Le Faouder and J. Bertrand-Michel from the MetaToul-lipidomic facility (Inserm UMR1048, Toulouse, France) and the assistance of A. Gaston and A. Estival from the Department of Pathology, IUCT-O for RNAscope analysis.

## Author contributions
C.I., A.M., M.F., E. March., F.B., C.C. designed and performed the experiments, collected and analyzed the data. E. Mart., J.G. and T.F. performed biostatistical analyses. M.M., O.B.S. and A.G.P. performed proteomic analyses. V.G. and S.C. performed the experiments. S.T.D. provided the Yumm cells. A.M., F.P., P.B., P.R., N.M. and L.L. collected, characterized, and analyzed human melanoma specimens. T.L. and B.S. provided expertize and edited the manuscript. N.A.A. and C.C. wrote the manuscript, supervised and coordinated the study.

## Competing interests
N.M. has worked as an investigator and/or consultant and/or speaker for BMS, MSD, Amgen, Roche, GSK, Novartis, Pierre Fabre. B.S. has worked as an investigator, consultant and speaker for BMS. The authors declare that they have no other conflict of interest.
