## [Peer Review File · Nature Communications]

Reviewers' comments:

Reviewer #1 (Remarks to the Author):

In this manuscript, the investigators show that sphingosine kinase-1 (SK1), which is overexpressed in many human tumors, regulates antitumor immunity. They report that lower SK1 expression correlates with longer survival upon PD1 blockade. They report that shSK1 tumor cells exhibit slow tumor growth in vivo with: 1) increased CD8+ tumor-infiltrating T cells (TILs) that produced higher IFN-g and upregulate a number of inhibitory receptors, and 2) decreased Tregs. They observe increased proliferation of CD8+ TILs in shSK1 tumors as compared to control tumors. They report that decreased SK1 expression goes along with decreased PGE2 synthetase expression. Finally, they report that high SK1 transcripts correlate with higher expression of a number of immune escape genes.

Overall, this study aims at demonstrating that SK1, which is known to be upregulated by multiple tumors and plays a role in tumor progression, may also modulate immune responses to melanoma and other tumors.

The findings as presented raised a number of major concerns that limit the interpretation of the findings. First, the experimental studies include little functional evaluation of tumor-infiltrating CD8+ T cells and Tregs, supporting the clinical effects of SK1 downregulation in tumors. Second, the mechanisms used by SK1 to impede immune responses to tumors remain unclear. Finally, it is difficult to reconcile the observation that increased SK1 expression in human tumors goes along with the upregulation of negative immunoregulatory markers that are usually found in the context of inflamed tumors that are more likely to respond to immune checkpoint blockade. It is also puzzling that many of these markers (Tim3, TIGIT...etc) appeared to increase in mouse tumors that downregulate SK1 as reported by the investigators (Figure 3).

MAJOR COMMENTS.

1. In Figure 1, additional info is needed to characterize primary melanoma and metastatic melanoma (Breslow, ulceration, site of metastasis).

The investigators would need to address the question of heterogeneity of SK1 expression in tumors.

The data in PD-1 treated patients are interesting although the cohort is small. One would like to know how the 50% cutoff has been determined and whether this goes along major differences in terms of T cell infiltration.

2. In figure 2 C, the decreased tumor growth of shSK1 Yumm1.7 is modest and one would like to see survival data.

3. In Figure 3A, the very high Ki67 expression by CD8+ T cells and Tregs in shcontrol tumors is extremely surprising. One would like to see gating strategy and negative controls. The authors report small differences in terms of IFN γ production, which cannot explain the difference in antitumor activity. One would like to see functional evidence supporting T-cell-mediated tumor reactivity and potentially lower Treg-mediated immunosuppression.

4. In Figure 4, the combinatorial therapies with aCTLA4 and aPD1 would need to include functional data of CD8+ T cells and Tregs. One would also like to see the expression of inhibitory ligands. The size of tumors at time of ICB should be indicated.

5. The observation of PGE2 synthetase downregulation interesting but its mechanism remains obscure

6. In Figure 6, the investigators report that high Sk1 expression in tumors correlates with the

upregulation of a number of inhibitory pathways known to be upregulated by T cells in inflamed tumors. This observation does not fit well with the previous observations made by the investigators that multiple inhibitor receptors (Tim3, TIGIT....etc) appeared to increase in mouse tumors that downregulate SK1 (Figure 3). The data in Figure 6 would rather suggest that high sk1 tumors are inflamed and may be more likely respond to immune checkpoint blockade. This observation is puzzling and it is unclear how this supports the main conclusion of the manuscript. In addition, it is unclear what the authors mean with "HIGH" SK1 expression.

Reviewer #2 (Remarks to the Author):

The MS entitled "Resistance of Melanoma to Immune Checkpoint Inhibitors is Overcome by targeting the Lipid Kinase SK1" from Caroline Imbert and colleagues studies the impact of the sphingosine kinase-1 (SK1) in the context of cancer immunotherapy. In the underlying MS the authors identified a novel resistance mechanism to ICI, which is timely and clinical relevant. Using preclinical models of cancer and melanoma patient data, they showed that downregulation of SK1 reduces tumour growth, improves ICI therapy and is associated with a favorable outcome of anti-PD1 therapy in melanoma patients, respectively. Despite this is a very interesting finding several concerns have been raised (see below). In general the study is very descriptive and lacks mechanistic insights. The authors attempt to gain some insights but these are not convincing in the current version. The translational aspect is questionable as the inhibitor used raises more questions than it delivers answers. The human data is descriptive and the statements made are speculative. This part needs a more careful characterization and more in depth analysis. All these points are accompanied by technical issues which need to be addressed to validate the initial findings. In summary, this study addresses an important topic and has potentially interesting and novel findings, which need to be substantiated in further experiments.

1. The authors should characterize their melanoma patient cohort. It is intriguing that SPHK1 high patients are less likely to respond to anti-PD1, however this could be explained by other parameters e.g. stage, tumour load, T cell infiltration etc. How is SPHK1 expressed across different melanoma stages? The data presented in Figure 1 is rather preliminary. Could it be that all good responder are IIIc IVa? Does SPHK1 correlate with T cell infiltration, maybe SPHK1 melanoma are T cell poor and thus do not respond? What about Treg infiltration. In the Figure legend the authors write n=11 SPHK1 high patients and n=19 SPHK1 low, however in table 1 there are only 22 patients annotated. Could SPHK1 be expressed by other cells in the tumour microenvironment? Which cells would express it? Overall the human data are interesting, but preliminary.
2. In figure 2. The authors show that downregulation of SK1, mediated by shRNA, reduces tumour growth in wild type mice. This effect is abolished in NSG and CD8-deficient mice. What are CD8-deficient mice? Nowhere in the MS the authors explain this mouse model. In general genetic alteration of CD8 or CD4 causes issues with T cell development (Immunology. 1996 Feb; 87(2):220-9.). The authors should perform antibody depletion experiments to dissect the role of CD8, CD4 and NK cells in this model.
3. Figure 3 is partially counter intuitive. Despite the increased proliferation and IFN γ production the authors describe increased expression of PD-1, CTLA4, TIM3. These are all exhaustion marker and associated with T cell dysfunction. Especially, when expressed at the same time by the same cell. This needs to be discussed in more detail (Blood 2011 117:4501-4510).
4. In Figure S4 the authors state SK1 silencing also abolished the ICI-induced increased expression of immune inhibitory checkpoint molecules... S4A However, there is no increase in CTLA4+PD1+ or PD1+TIM3+ CD8 t cells upon aCTLA4 or aPD1 treatment. What do the authors mean?
5. The therapeutic effect of aCTLA4 and PD1 is impressive. However, the mechanistic insights are limited. Treating shSK1 tumours with aCTLA4 only slightly increases the number of tumour infiltrating CD8 T cells and marginally reduces the number of Treg cells. This slight changes hardly explain the dramatic phenotype. The statistical comparisons are also misleading. The authors should compare frequency of CD8 and Tregs between Ctrl shSK1 and a CTLA4 shSK1!!! This does not look significant!!!

6. The authors write on p. 8 SKI-I treatment also enhanced anti-PD-1 efficacy (Fig 4G)...although statistically significant, there is no biological meaningful effect of SKI-I in combination with anti-PD1. This sentence should be rephrased. On a different note, these data are actually contradicting the findings present above where shRNA-mediated knockdown in combination with PD-1 leads to the cure of 2/3 of mice. The explanation the inhibitor is not as effective is not helpful, as this inhibitor shows efficacy in combination with CTLA-4 (Fig 4F). It is also to mention that the knockdown of SHPK1 is incomplete!

7. In Figure 4 H the author use MC38. It is surprising that anti-PD1 treatment has only a marginal therapeutic effect, given that this cell line is one of the most sensitive models.

8. Can the authors by re-expressing SK1 in Yumm cells revert the observed phenotype? This experiment would rule out off-target effect (well known for shRNAs) and strengthen the findings presented so far.

9. In Figure 5 the authors attempt to unravel the molecular mechanisms causing the profound phenotype described before. For this, they have used a Proteomic screen. In this screen, the authors found decreased expression of Ptges. However, this could be a shRNA off-target effect. This concern is substantiate by the fact that there is a dramatic difference of Ptges expression between shSK11 and shSK2, although the have similar SK1 expression. Again this problem should be addressed, by 1. Re-expressing SK1 to see whether this restores Ptges expression and 2. by analyzing the CRISPR-KO cells, which are less likely to have off-targets compared to shRNA.

10. The correlation data presented in Figure 6. are interesting for PTGES, FoxP3, TGFB1 and CTLA4. Although significant, the correlation coefficient is very low for the other marker. These data also raises the question whether Tregs express PTGES and/or SK1?

Minor comments:

In the MM section under Tumor Challenge and treatments: there is no explanation what CD8-deficient mice are.

In the MM section the authors state "SK1 activity was determined as described 52 with minor modifications." Either the modifications are important to reproduce the findings, than they should be explained! Or it is not important than please delete.

Response to Reviewers' Comments:

The point-by-point response to the reviewers' comments is given in blue and in bold.

Reviewer #1 (Remarks to the Author):

In this manuscript, the investigators show that sphingosine kinase-1 (SK1), which is overexpressed in many human tumors, regulates antitumor immunity. They report that lower SK1 expression correlates with longer survival upon PD1 blockade. They report that shSK1 tumor cells exhibit slow tumor growth in vivo with: 1) increased CD8+ tumor-infiltrating T cells (TILs) that produced higher IFN-g and upregulate a number of inhibitory receptors, and 2) decreased Tregs. They observe increased proliferation of CD8+ TILs in shSK1 tumors as compared to control tumors. They report that decreased SK1 expression goes along with decreased PGE2 synthetase expression. Finally, they report that high SK1 transcripts correlate with higher expression of a number of immune escape genes.

Overall, this study aims at demonstrating that SK1, which is known to be upregulated by multiple tumors and plays a role in tumor progression, may also modulate immune responses to melanoma and other tumors. The findings as presented raised a number of major concerns that limit the interpretation of the findings. First, the experimental studies include little functional evaluation of tumor-infiltrating CD8+ T cells and Tregs, supporting the clinical effects of SK1 downregulation in tumors. Second, the mechanisms used by SK1 to impede immune responses to tumors remain unclear. Finally, it is difficult to reconcile the observation that increased SK1 expression in human tumors goes along with the upregulation of negative immunoregulatory markers that are usually found in the context of inflamed tumors that are more likely to respond to immune checkpoint blockade. It is also puzzling that many of these markers (Tim3, TIGIT....etc) appeared to increase in mouse tumors that downregulate SK1 as reported by the investigators (Figure 3).

MAJOR COMMENTS.

1. In Figure 1, additional info is needed to characterize primary melanoma and metastatic melanoma (Breslow, ulceration, site of metastasis).

1.A. As requested, additional info concerning clinical data have been added for Figure 1 (Supplementary Table 1 and 2; Table 1). Please note that Breslow thickness and signs of ulceration are not always reported in metastatic melanoma patients from the anti-PD-1 cohort (Table 1).

The investigators would need to address the question of heterogeneity of SK1 expression in tumors.

1.B. One reason explaining the difficulty to treat melanoma is related to the heterogeneity of tumors. The source of this heterogeneity is unclear, however, it is now accepted that primary and metastatic lesions are composed of a mixture of cancer cells harboring a proliferative or an invasive phenotype (1, 2). Under microenvironmental stress, melanoma cells change their transcription programs to switch back-and-forth between proliferative and invasive states. For instance, cancer cells adapt to the hypoxic microenvironment by regulating the expression of hypoxia-inducible factor (HIF) family of transcription factors and HIF-target genes. In human cutaneous melanoma biopsies, changes in the expression levels of HIF as well as melanocytic markers, such as the master regulator of melanocyte differentiation MITF, were associated with the presence of hypoxic regions (3). As a matter of fact, HIF-2 α directly regulates the expression of *SPHK1* by binding to hypoxia response elements (HRE) in the *SPHK1* proximal promoter (4). Moreover, we recently showed that MITF expression controlled the ceramide metabolism during the phenotypic plasticity phenomena (5). Although, we agree with the reviewer the question of heterogeneity of SK1 expression in tumors is of interest, we consider this issue goes beyond the scope of the paper.

The data in PD-1 treated patients are interesting although the cohort is small. One would like to know how the 50% cutoff has been determined and whether this goes along major differences in terms of T cell infiltration.

1.C. We performed new RNAscope staining for *SPHK1* mRNA using a second cohort of melanoma patients treated with anti-PD-1 (Nivolumab or Pembrolizumab). New survival data are described in the updated figure 1 and now include a cohort of 32 patients. Kaplan-Meier estimates were used to generate PFS and OS curves for patients with high or low tumor cells positive for *SPHK1* expression (≤ 50 and $>50\%$) (p-value equal to 0.0112 for PFS and 0.0445 for OS). This cutoff has been arbitrarily determined in order to discriminate between poor and good responders. We also calculated the p-

value for PFS and OS with the median value ($\leq 30\%$ or $>30\%$ of tumor cells positive for *SPHK1* staining). We obtained statistical significance value for PFS ($p=0.025$) but not for OS (0.1305).

To evaluate the CD8 T cell and Treg infiltration in tumors, we also performed IHC staining and used the CD8 and Foxp3 proteins as markers. Analyses were then completed by performing PD-L1 staining. These biomarkers are often used to stratify patients as clinical responders or non-responders to anti-PD-1 therapy. In line with our preclinical data, *SPHK1* expression was significantly associated with Foxp3 staining, however no association was found with CD8 or PD-L1 staining (see table below). Of note, Foxp3 staining was very low and the majority of Foxp3 negative or weak tumors fell in the *SPHK1* low group. Although we find these data to be really interesting, the low level of Foxp3 staining led us to be cautious and choose not to depict these data in the paper.

	Total N=32	SPHK1low ($\leq 50\%$) N= 21	SPHK1high ($>50\%$) N= 11
Foxp3 score (n=32)			p = 0.0367
negative or weak	27 (84.4%)	20 (95.2%)	7 (63.6%)
positive ($>1\%$)	5 (15.6%)	1 (4.8%)	4 (36.4%)
CD8 score (n=29)			p = 0.3568
absent or no brisk	22 (75.9%)	17 (81.0%)	5 (62.5%)
brisk	7 (24.1%)	4 (19.0%)	3 (37.5%)
Missing	3	0	3
PD-L1 score(n=29)			p = 1.0000
negative	14 (48.3%)	10 (47.6%)	4 (50.0%)
positive ($>1\%$)	15 (51.7%)	11 (52.4%)	4 (50.0%)
Missing	3	0	3

Table 2. Foxp3, CD8 and PD-L1 expression in tumor samples exhibiting low and high *SPHK1* expression. The chi-squared or Fisher's exact test was used to compare categorical variables.

2. In figure 2 C, the decreased tumor growth of shSK1 Yumm1.7 is modest and one would like to see survival data.

3. In Figure 3A, the very high Ki67 expression by CD8+ T cells and Tregs in shctrl tumors is extremely surprising.

- One would like to see gating strategy and negative controls.

The gating strategies and the isotype control for KI67 in tumor and in tumor draining lymph nodes (TDLN) are described below. Similar profiles were observed in several articles (6, 7)

Representative flow cytometry gating strategy for T cells. Flow cytometric analysis of KI67 positive CD4 and CD8 T cells in tumors (TUM) and tumor draining lymph node (TDLN). Iso: Isotype control

The authors report small differences in terms of IFN γ production, which cannot explain the difference in antitumor activity.

- One would like to see functional evidence supporting T-cell-mediated tumor reactivity and potentially lower Treg-mediated immunosuppression.

As requested by the Reviewer, we analyzed the antitumor activity of tumor-infiltrating CD8 T cells, and we observed a higher cytolytic activity as measured by expression of CD107a (LAMP-1) and granzyme B (new Supplementary Fig. 1).

SK1 downregulation enhances the anti-tumor activity of tumor-infiltrating CD8 T cells. Antitumor activity of tumor-infiltrating CD8 T cells as measured by the expression of IFN-γ, LAMP-1 (CD107a) and granzyme B (Grz) at day 11. These data are depicted in the new supplementary Fig.1

We also tried to analyze the suppressive activity of Treg from shCtrl or shSK1 tumors using DERE (Foxp3-DTR-EFGP) mice at day 11. However, we failed to obtain a sufficient amount of tumor-infiltrating Treg cells to perform the suppressive assay. Nevertheless, we performed a suppressive activity test using Treg isolated from the tumor draining lymph nodes of mice grafted with shCtrl and shSK1 melanoma cells and found no significant differences as shown in the figure below.

4. In Figure 4, the combinatorial therapies with aCTLA4 and aPD1 would need to include functional data of CD8+ T cells and Tregs.

4.A. As requested the Reviewer, we analyzed the functional activity of tumor-infiltrating CD8+ T cells after combinatorial therapies. We observed that the frequency of IFN- γ CD8 positive cells was increased in SK1 silenced tumors as compared to shCtrl ones. This phenomenon was also observed in the context of ICI therapy (new supplementary figure 4B). In contrast, we observed a decrease in the frequency of TNF+ CD8+ TILs, which likely reflect a partial dysfunction of CD8+ T cells in SK1 knockdown tumors. As a matter of fact, anti-CTLA-4 or anti-PD-1 treatment increased the proportion of TNF+ CD8+ TILs, indicating that ICI may restore some biological functions of TILs, leading to tumor rejection (new supplementary figure 4B).

In addition, SK1 silencing increased the proportion of PD-1+ CD8+ TILs co-expressing CTLA-4 or TIM-3 at day 11 post-melanoma cell injection. Whereas the co-expression of PD-1 with CTLA-4 or TIM-3 likely reflects the activation of TILs at early time points, those immune checkpoints trigger TIL dysfunctions at later stages, facilitating immune escape and melanoma progression (8, 9). Interestingly, anti-CTLA-4 or anti-PD-1 prevented the increase of CTLA-4 and TIM-3 on PD-1+ CD8+ TILs in SK1 knockdown tumors (new Fig. 4). Moreover, we also observed an upregulation of CD226 and a downregulation of TIGIT on CD8+ and CD4+ TILs upon anti-PD-1 in SK1 knockdown tumors (supplementary Fig. 4C and D). Together, our findings indicate that the synergism between SK1 knockdown and ICI therapy depends, at least in part, on the modulation of immune checkpoint expression on TILs.

We added in the new figure 4 the percentage of Foxp3+ cells among total tumor cells (Fig. 4D), indicating that anti-CTLA-4 or anti-PD-1 therapy further decreases Treg content in SK1 knockdown tumors. This phenomenon is associated with a decrease expression of CCL17 and CCL22 in shSK1 tumors leading to a lower accumulation of Treg in the tumor (Fig. 4D), irrespectively of ICI treatment.

- One would also like to see the expression of inhibitory ligands.

4.B. As suggested by the reviewer, additional experiments have been performed to address this issue. We analyzed the expression of PD-L1, PD-L2 and Galectin-9 in tumors at day 11 on CD45 negative cells and myeloid cells: CD11b+Gr1+ (MDSC) and CD11b+Gr1-F480+ (TAM). We found a significant decrease for PD-L1 expression (but not for PD-L2) on CD45- cells in shSK1 tumors with and without ICI therapy. These results are now included in the revised version in the supplementary figure 5. As regard to galectin-9, its expression on MDSC, TAM and CD45- cells was weak and seemed slightly reduced in shSK1 tumors. We however decided not to depict these last result in the revised version.

A**Expression of inhibitory ligands after SK1 silencing.**

(A). PD-L1, PDL-2 and Galectin 9 expression on CD45-, CD11b+GR1+ (MDSC), CD11b+Gr1-F480+ (TAM) cells. (B). PD-L1 expression in CD45- cells from in tumors at day 11 after injection of shCtrl or shSK1(1) Yumm cells. (C) PD-L1 expression in CD45- cells from in tumors at day 11 after injection of shCtrl(2) or shSK1(3) Yumm cells and treatment with anti-CTLA-4 or anti-PD-1.

B**C**
- The size of tumors at time of ICB should be indicated.

Anti-PD-1 and anti-CTLA-4 were injected on day 5, 7 and 10 and 7, 10 and 13, respectively. As requested by the Reviewer the tumor volumes were 43.50 mm³ (shCtrl melanoma) versus 152.9 mm³ (shSK1 melanoma) at day 7 (n=10-12).

5. The observation of PGE2 synthetase downregulation interesting but its mechanism remains obscure Several potential consensus binding sites for transcription factors, like GC boxes, Barbie boxes and an aryl hydrocarbon regulatory element (ARE) have been identified in the promoter region of *PTGES/mPGES1* (10). During inflammation, a number of studies have reported that EGR1 promotes *PTGES* expression by itself (11) or coordinately with NF- κ B (12) in a variety of tumor cell lines. Of note, SK1 has been shown to stimulate NF- κ B activation in melanoma cell lines (11). Moreover, SK1 down-

regulation using siRNA was shown to reduce *PTGES* expression via a decrease of *EGR1* expression in HUVECs (13).

In this context, we analyzed *Egr1* expression by RT-qPCR in Yumm melanoma cells in which SK1 was downregulated or not. Although *Ptges* mRNA expression levels were reduced by SK1 silencing, those of *Egr1* were not affected (see the new supplementary Fig.10A and Fig below). In accordance with this observation, the re-expression of SK1 in shSK1 melanoma cells restored *Ptges* expression without affecting that of *Egr1*. These data demonstrate that SK1 regulates *Ptges* expression in an *Egr1*-independent manner in murine melanoma cells. However, *EGR1* mRNA expression was shown to be highly regulated by EGR1 itself, NF- κ B, serum response elements or AP-1 in human cells. It is also dependent on acetylation mechanisms (14). The multilevel complexity linking SK1 to *Ptges* regulation is currently under investigation in our laboratory and will be the subject of a separate study.

SK1 regulates *Ptges* expression. Relative expression levels for *Sphk1* (A), *Ptges* (B) and *Egr1* (C) expression were measured in shCtrl(2) or shSK1(3) Yumm cells transfected with an empty plasmid (mock) or a plasmid encoding murine SK1 (mSK1) (n=8-9). Data were compared using Kruskal-Wallis test with Dunn's correction.

6. In Figure 6, the investigators report that high Sk1 expression in tumors correlates with the upregulation of a number of inhibitory pathways known to be upregulated by T cells in inflamed tumors. This observation does not fit well with the previous observations made by the investigators that multiple inhibitor receptors (Tim3, TIGIT....etc) appeared to increase in mouse tumors that downregulate SK1 (Figure 3).

We observed an upregulation of CTLA-4, PD-1 and TIM-3 expression in CD8 T cells (Fig. 3, and sup. Fig. 4) and no changes were observed for TIGIT expression (sup. Fig.4) after SK1 silencing at day 11. At this time point, these upregulations likely reflect a higher activation phenotype associated with an increased proliferation (KI67+, Fig.3) and antitumor activity (IFN γ , GrzB, CD107a, supplementary Fig. 1C), rather than an exhaustion phenotype. Moreover, these three checkpoint molecules are upregulated upon T cell activation during the anti-tumor response, and tumor-infiltrating CTLA-4+ PD-1+ CD8+ T cells have been shown to contain the majority of tumor-antigen specific T cells (15, 16). We performed additional experiments to address this very important point and analyzed the proportion of PD-1+ Eomes+ CD8+ T cells that have been defined as an exhausted T cell populations in chronic infection and in melanoma (17, 18). Although, we did not detect any Eomes+ CD8+ T cells at day 11, we saw a decreased frequency of PD-1+ Eomes+ and PD-1+ TIM3+ CD8+ T cells at day 23 in shSK1 tumors that could reflect a decreased exhaustion phenotype in these tumors at later time points (8, 9). Since this tenet remains to be confirmed by additional experiments, we prefer not to depict these data in the revised version.

CD8/Treg ratio and percentage of PD-1+TIM-3+ and PD-1+Eomes+ CD8 TILs at day 23. shCtrl(2) or shSK1(3) Yumm melanoma cells were injected intradermally in C57BL/6 mice, and TIL content was analyzed by flow cytometry on day 23.

The data in Figure 6 would rather suggest that high sk1 tumors are inflamed and may be more likely respond to immune checkpoint blockade. This observation is puzzling and it is unclear how this supports the main conclusion of the manuscript. In addition, it is unclear what the authors mean with “HIGH” SK1 expression.

We agree with the Reviewer that one of the limitations to the successful use of ICI to treat cancer patients is the requirement for tumors to be pre-infiltrated with immune cells (i.e., inflamed tumors) and more specifically TILs. However, a high proportion of patients affected by cancers that are described as “immunogenic” still poorly respond to these therapies. As opposed to the “primary resistance”, these resistance mechanisms are defined as “adaptive immune resistance” and “acquired resistance” (19).

In figure 6B, we observed an interesting correlation between *SPHK1* expression and *PTGES* or *TGFB1*. Increased level of TGFβ is associated with poor prognosis in multiple different tumor types (20, 21), Moreover, preclinical models have shown that TGF-β receptor kinase inhibitor I synergizes with anti-CTLA-4 to inhibit tumor growth in the BRAFV600E PTEN^{-/-} melanoma model (Hanks et al., 2014). Pges produces PGE2, a major mediator of resistance to immunotherapies, including oncolytic vaccinia virotherapy (22). COX2 inhibitors, used to block PGE2 production, have been used to improve the efficacy of immunotherapy (23). Furthermore, *Ptges* has been described as a regulator of immunosuppression in cutaneous melanoma (24). Therefore, the data depicted in Figure 6 further argue that high SK1 tumors are likely more resistant to immune checkpoint therapy, in accordance with our data depicted in Fig. 1.

High *SPHK1* patients from TCGA metastatic melanoma cohort are in the 80th percentile as regard to *SPHK1* expression in melanoma samples. Low *SPHK1* patients are in the 20th percentile.

Reviewer #2 (Remarks to the Author):

The MS entitled “Resistance of Melanoma to Immune Checkpoint Inhibitors is Overcome by targeting the Lipid Kinase SK1” from Caroline Imbert and colleagues studies the impact of the sphingosine kinase-1 (SK1) in the context of cancer immunotherapy. In the underlying MS the authors identified a novel resistance mechanism to ICI, which is timely and clinical relevant. Using preclinical models of cancer and melanoma patient data, they showed that downregulation of SK1 reduces tumour growth, improves ICI therapy and is associated with a favorable outcome of anti-PD1 therapy in melanoma patients, respectively. Despite this is a very interesting finding several concerns have been raised (see below). In general the study is very descriptive and lacks mechanistic insights. The authors attempt to gain some insights but these are not convincing in the current version. The translational aspect is questionable as the inhibitor used raises more questions than it delivers answers. The human data is descriptive and the statements made are speculative. This part needs a more careful characterization and more in depth analysis. All these points are accompanied by technical issues which need to be addressed to validate the initial findings. In summary, this study addresses an important topic and has potentially interesting and novel findings, which need to be substantiated in further experiments.

MAJOR COMMENTS.

1. The authors should characterize their melanoma patient cohort. It is intriguing that SPHK1 high patients are less likely to respond to anti-PD1, however this could be explained by other parameters e.g. stage, tumour load, T cell infiltration etc. How is SPHK1 expressed across different melanoma stages?

The clinical characteristics of melanoma patients in our cohort is described in the new Table 1. There are no differences in *SPHK1* high and *SPHK1* low group for tumor stage, histological subtype, and mutation status (BRAF and NRAS). Concerning the level of T cell infiltrate observed in these tumors, please see response to question 1c from Reviewer 1.

The data presented in Figure 1 is rather preliminary. Could it be that all good responder are IIIc IVa? Does SPHK1 correlate with T cell infiltration, maybe SPHK1 melanoma are T cell poor and thus do not respond? What about Treg infiltration. In the Figure legend the authors write n=11 SPHK1 high patients and n=19 SPHK1 low, however in table 1 there are only 22 patients annotated. Could SPHK1 be expressed by other cells in the tumour microenvironment? Which cells would express it? Overall the human data are interesting, but preliminary.

We performed new RNAscope staining for *SPHK1* mRNA using a second cohort of melanoma patients treated with anti-PD-1 (Nivolumab or Pembrolizumab). Therefore, we obtained new survival data described in the new figure 1 with 32 patients (Low *SPHK1* group, n=21; High *SPHK1* group, n=11). For illustration purposes, Kaplan-Meier estimates were used to produce PFS and OS curves with two groups (≤ 50 and $>50\%$) (p-value equal to 0.0112 for PFS and 0.0445 for OS). The error in the figure legend has been corrected accordingly.

Moreover, we analyzed T cell infiltration by IHC staining for CD8, Foxp3 and PD-L1. Although not included in the current manuscript, we found that the longer progression-free survival and overall survival found in the Low *SPHK1* group could not be explained by differences on CD8 and PD-L1 staining on tumor biopsies. However, the Foxp3 staining was positively associated with the *SPHK1* expression, in line with our preclinical data. Since the Foxp3 staining was low, we prefer not to depict those data in the paper.

Additionally, as reviewer 2 rightfully mentioned, *SPHK1* is expressed by other cells from the tumor microenvironment, the main subtype being endothelial cells, which constituted one of our quality control criteria for the RNAscope staining.

As mentioned in our response to Reviewer 1's question 1c, we performed new analysis and stained FFPE tumor sections from melanoma patients for the detection of CD8 and PD-L1 markers. One important observation was that CD8 (bright versus no bright), PD-L1 (positive versus negative or $<1\%$) or Foxp3 (positive versus negative) markers did not allow to discriminate between good responders versus non-responders for anti-PD-1 therapy using Kaplan Meier analysis (see figure below). Under these settings, *SPHK1* staining seems to be a more powerful variable to predict response of patients to anti-PD1 treatment.

2. In figure 2. The authors show that downregulation of SK1, mediated by shRNA, reduces tumour growth in wild type mice. This effect is abolished in NSG and CD8-deficient mice. What are CD8-deficient mice? Nowhere in the MS the authors explain this mouse model. In general genetic alteration of CD8 or CD4 causes issues with T cell development (Immunology. 1996 Feb;87(2):220-9.). The authors should perform antibody depletion experiments to dissect the role of CD8, CD4 and NK cells in this model.

As requested by the Reviewer, we now provide more information regarding CD8 deficient mice in the methods section of the paper. We also performed *in vivo* depletion experiments and we demonstrated that CD4 and CD8 cells, but not NK cells, are important lymphocytes in tumor growth inhibition following SK1 silencing.

These results depicted below are now included in the revised version of the manuscript in supplementary Fig.3.

In vivo depletion of NK, CD4 and CD8 cells. Mice received an intraperitoneal injection of anti-CD4, anti-CD8, and anti-NK1 one day before intradermal injection of shCtrl(2) or shSK1(3) Yumm cells, two days after and then every six days.

3. Figure 3 is partially counter intuitive. Despite the increased proliferation and IFN γ production the authors describe increased expression of PD-1, CTLA4, TIM3. These are all exhaustion marker and associated with T cell dysfunction. Especially, when expressed at the same time by the same cell. This needs to be discussed in more detail (Blood 2011 117:4501-4510).

At day 11, we observed an upregulation of CTLA-4, PD-1 and TIM-3 expression but not TIGIT expression on CD8+ TILs (Fig. 3, and sup. Fig. 4) following SK1 silencing. At this time point, this upregulation seems likely to reflect a higher activation phenotype associated with an increased proliferation (KI67+) of TILs as depicted in Fig. 3 and antitumor activity (IFN γ , GrzB, CD107a, supplementary Fig. 1C), rather than an exhaustion phenotype. Moreover, these three checkpoint molecules are upregulated upon T cell activation during the anti-tumor response, and tumor-infiltrating CTLA-4+ PD-1+ CD8+ T cells have been shown to contain the majority of tumor-antigen specific T cells (15, 16).

We performed additional experiments to address this very important point in order to analyze the expression of PD-1+ Eomes+ CD8 T cells that have been defined as an exhausted T cells in chronic infection and in melanoma (17, 18). Whereas we didn't observe Eomes+ CD8+ T cells at day 11, we observed a decreased frequency of PD-1+ Eomes+ and PD-1+ TIM3+ CD8 T cells at day 23 in shSK1 group that could reflect a decreased exhaustion phenotype in SK1 knockdown tumors at late time points (8, 9). Since this tenet remains to be confirmed by additional experiments, we prefer not to depict those data.

CD8/Treg ratio and percentage of PD-1+TIM-3+ and PD-1+Eomes+ CD8 TILs at day 23. shCtrl(2) or shSK1(3) Yumm melanoma cells were injected intradermally in C57BL/6 mice, and TIL content was analyzed by flow cytometry on day 23.

4. In Figure S4 the authors state SKI silencing also abolished the ICI-induced increased expression of immune inhibitory checkpoint molecules... S4A However, there is no increase in CTLA4+PD1+ or PD1+TIM3+ CD8 t cells upon aCTLA4 or aPD1 treatment. What do the authors mean?

We thank the Reviewer for raising this issue that was not properly explain in the previous version of our manuscript. We observed an increase of CTLA-4, PD-1 and TIM-3 after SK1 silencing and without ICI treatment.

SK1 knockdown increased the proportion of PD-1+ CD8+ TILs co-expressing CTLA-4 or TIM-3 at day 11 post-melanoma cell injection. Although, the co-expression of PD-1 with CTLA-4 or TIM-3 likely reflects the activation of TILs at early time points, those immune checkpoints trigger TIL dysfunctions at later stages, facilitating immune escape and melanoma progression (25, 26). Interestingly, anti-CTLA-4 or anti-PD-1 prevented this increase of CTLA-4 and TIM-3 molecules on PD-1+ CD8+ TILs in SK1 knockdown tumors (new Fig. 4). Moreover, we also observed an upregulation of CD226 and a downregulation of TIGIT on CD8+ and CD4+ TILs upon anti-PD-1 in SK1 knockdown tumors (supplementary Fig. 4C and D). Together, our findings indicate that the synergism between SK1 knockdown and ICI therapy depends, at least in part, on the modulation of immune checkpoint expression on TILs.

5. The therapeutic effect of aCTLA4 and PD1 is impressive. However, the mechanistic insights are limited. Treating shSK1 tumours with aCTLA4 only slightly increases the number of tumour infiltrating CD8 T cells and marginally reduces the number of Treg cells. This slight changes hardly explain the dramatic phenotype. The statistical comparisons are also misleading. The authors should compare frequency of CD8 and Tregs between Ctrl shSK1 and a CTLA4 shSK1!!! This does not look significant!!!

As mentioned in our response to Reviewer 1's question 4, we analyzed the functional activity of tumor-infiltrating CD8+ T cells after combinatorial therapies. We observed that the frequency of IFN- γ CD8 positive cells was increased in shSK1 tumors as compared to shCtrl tumors. This phenomenon was also observed in the context of ICI therapy (new supplementary figure 4B). In contrast, we observed a decreased in the frequency of TNF+ CD8+ TILs, which likely reflect a partial dysfunction of CD8+ T cells in SK1 knockdown tumors. As a matter of fact, anti-CTLA-4 or anti-PD-1 injection increased the proportion of TNF+ CD8+ TILs, indicating that ICI may restore some biological functions of TILs, leading to tumor rejection (new supplementary figure 4B).

We added in the new figure 4 the percentage of Foxp3+ cells among total tumor cells (Fig. 4D), indicating that anti-CTLA-4 or anti-PD-1 therapy further decreases Treg content in SK1 knockdown tumors. This phenomenon is associated with a decreased expression of CCL17 and CCL22 in shSK1 tumors leading to a lower accumulation of Treg in the tumors (Fig. 4D), irrespectively of ICI treatment.

6. The authors write on p. 8 SKI-I treatment also enhanced anti-PD-1 efficacy (Fig 4G)...although statistically significant, there is no biological meaningful effect of SKI-I in combination with anti-PD1. This sentence should be rephrased. On a different note, these data are actually contradicting the findings present above where shRNA-mediated knockdown in combination with PD-1 leads to the cure of 2/3 of mice. The explanation the inhibitor is not as effective is not helpful, as this inhibitor shows efficacy in combination with CTLA-4 (Fig 4F). It is also to mention that the knockdown of SHPK1 is incomplete!

In accordance with the results obtained with the knockdown strategy (Fig. 4A), the pharmacological approach showed that SK1 inhibition sensitizes melanoma tumors more efficiently to anti-CTLA-4 than anti-PD-1 treatment (Fig. 4G). However, as aptly mentioned by the Reviewer, the effects of SKI-I were less pronounced as compared to SK1 knockdown for tumors treated with anti-PD-1. One could speculate that, under our experimental conditions, SKI-I has also affected other SK1-dependent signaling pathways in host cells from the tumor microenvironment that have limited the impact of anti-PD-1 and, to a lesser extent, anti-CTLA-4. However, we cannot exclude that SK1 inhibition using SKI-I *in vivo* is not as powerful as when using the shRNA approach and careful additional studies in mice harboring melanoma would be required to determine the best treatment conditions (dose, method of administration, pharmacokinetics, and chemical nature of the inhibitor). Nonetheless, our data do show the potential of such pharmacological approach to improve response to ICI in mouse melanoma. As requested by the Reviewer, the sentence (page 9 of the revised version) has been modified as follow: "In addition, albeit less potently than SK1 silencing, pharmacological inhibition of SK1 by SKI-I significantly improved the anti-CTLA-4 therapy on established Yumm melanoma, and led to a potent increase in tumor rejection and animal survival (Fig. 4F). Under these conditions, SKI-I treatment slightly enhanced the anti-PD-1 efficacy (Supplementary Fig. 7A). Interestingly, the potentiation of anti-PD-1 efficacy was more pronounced in the MC38 colon cancer model (Supplementary Fig. 7B).

7. In Figure 4 H the author use MC38. It is surprising that anti-PD1 treatment has only a marginal therapeutic effect, given that this cell line is one of the most sensitive models.

We fully agree with reviewer that MC38 are sensitive to anti-PD-1 therapy. To be able to assess a potential benefit of combining the SK1 inhibitor to anti-PD-1 treatment, in this model we optimized our experiment settings and injected only 100 µg per mouse of anti-PD-1 at day 7 and 10 (as opposed to the 200 µg per mouse used in the literature). This allowed us to observe a moderate response to anti-PD1 therapy alone and provided us with a good model to study the impact of SKI-I in these settings. We could obtain a higher therapeutic effect with effect with 3 injections (at day 7, 10 and 13) of 200 µg per mice of anti-PD-1.

8. Can the authors by re-expressing SK1 in Yumm cells revert the observed phenotype? This experiment would rule out off-target effect (well known for shRNAs) and strengthen the findings presented so far.

As suggested by the Reviewer we overexpressed SK1 in shSK1 Yumm cells by transfecting them with a plasmid expressing the mouse SK1 cDNA. SK1 re-expression reverted the impact on tumor growth, as described in the new supplementary figure 10 A.

Re-expressing SK1 restores tumor growth and Ptges expression. shCtrl(2) + mock or shSK1(3) + mock or shSK1(3) + mSK1 Yumm cells were injected in wild-type mice. Tumor volumes, presented as means ± SEM (n=9-12). Samples were compared using two-way ANOVA test.

9. In Figure 5 the authors attempt to unravel the molecular mechanisms causing the profound phenotype described before. For this, they have used a Proteomic screen. In this screen, the authors found decreased expression of Ptges. However, this could be a shRNA off-target effect. This concern is substantiate by the fact that there is a dramatic difference of Ptges expression between shSK11 and shSK2, although the have similar SK1 expression. Again this problem should be addressed, by :1. Re-expressing SK1 to see whether this restores Ptges expression and 2. by analyzing the CRISPR-KO cells, which are less likely to have off-targets compared to shRNA.

We agree with the Reviewer that it is necessary to demonstrate that the inhibitory effect of the shRNA against *Sphk1* on the expression of *Ptges* is not an off-target effect. Establishment of a full SK1 KO Yumm cell line, using the CRISPR/Cas9 technology, is not possible as the deletion of SK1 highly alters cell proliferation. Of note, supplementary Fig. 1 depicting B16 melanoma cells for which the CRISPR technology was used to inhibit SK1 expression is not a full KO. Residual SK1 expression was found in this cell line to levels comparable to the ones observed when using the shRNA technology. As this figure may be misleading we choose to remove it from the manuscript and address the potential off-target effect using a third shRNA targeting a different sequence than the first two previously used in the manuscript (see figure below).

As shown in supplementary figure 1A and 10B, shSK1(3) cells exhibited a markedly reduced expression of SK1. As expected, silencing of SK1 was associated with a decreased expression of *Ptges*.

Moreover, to completely exclude off-target effect, we performed a rescue experiment by transfecting shSK1(3) cells with a plasmid encoding murine SK1 (+ mSK1). Under these conditions, the inhibitory effect on *Ptges* expression was totally abolished by the rescue of SK1 expression. These results are now included in new supplementary Fig. 10. Thus, all this additional data strongly supports the reproducibility of the molecular mechanism.

10. The correlation data presented in Figure 6. are interesting for PTGES, FoxP3, TGFB1 and CTLA4. Although significant, the correlation coefficient is very low for the other marker. These data also raises the question whether Tregs express PTGES and/or SK1?

We compared the expression of *Ptges* and *Sphk1* in Treg using transcriptomic data set obtained from murine T cells (27). The data from 3 replicates are described below. *Ptges* and *Sphk1* have similar expression in Treg and conventional T cells (Tconv) from thymus and lymph nodes.

Minor comments:

In the MM section under Tumor Challenge and treatments: there is no explanation what CD8-deficient mice are.

We apologize for this omission, CD8-deficient mice are CD8α Knockout mice and have been kindly given by Prof. J. van Meerwijk, INSERM U1043, Toulouse, France.

In the MM section the authors state “SK1 activity was determined as described 52 with minor modifications.” Either the modifications are important to reproduce the findings, than they should be explained! Or it is not important than please delete.

We agree with the reviewer and we decided to delete this sentence.

References

1. Ahmed F, and Haass NK. Microenvironment-Driven Dynamic Heterogeneity and Phenotypic Plasticity as a Mechanism of Melanoma Therapy Resistance. *Front Oncol.* 2018;8(173).
2. Hoek KS, Eichhoff OM, Schlegel NC, Dobbeling U, Kobert N, Schaerer L, Hemmi S, and Dummer R. In vivo switching of human melanoma cells between proliferative and invasive states. *Cancer Res.* 2008;68(3):650-6.
3. Widmer DS, Hoek KS, Cheng PF, Eichhoff OM, Biedermann T, Raaijmakers MIG, Hemmi S, Dummer R, and Levesque MP. Hypoxia contributes to melanoma heterogeneity by triggering HIF1alpha-dependent phenotype switching. *J Invest Dermatol.* 2013;133(10):2436-43.
4. Anelli V, Gault CR, Cheng AB, and Obeid LM. Sphingosine kinase 1 is up-regulated during hypoxia in U87MG glioma cells. Role of hypoxia-inducible factors 1 and 2. *J Biol Chem.* 2008;283(6):3365-75.
5. Leclerc J, Garandeau D, Pandiani C, Gaudel C, Bille K, Nottet N, Garcia V, Colosetti P, Pagnotta S, Bahadoran P, et al. Lysosomal acid ceramidase ASAH1 controls the transition between invasive and proliferative phenotype in melanoma cells. *Oncogene.* 2019;38(8):1282-95.
6. Juneja VR, McGuire KA, Manguso RT, LaFleur MW, Collins N, Haining WN, Freeman GJ, and Sharpe AH. PD-L1 on tumor cells is sufficient for immune evasion in immunogenic tumors and inhibits CD8 T cell cytotoxicity. *J Exp Med.* 2017;214(4):895-904.
7. Redmond WL, Linch SN, and Kasiewicz MJ. Combined targeting of costimulatory (OX40) and coinhibitory (CTLA-4) pathways elicits potent effector T cells capable of driving robust antitumor immunity. *Cancer Immunol Res.* 2014;2(2):142-53.
8. Koyama S, Akbay EA, Li YY, Herter-Sprue GS, Buczkowski KA, Richards WG, Gandhi L, Redig AJ, Rodig SJ, Asahina H, et al. Adaptive resistance to therapeutic PD-1 blockade is associated with upregulation of alternative immune checkpoints. *Nat Commun.* 2016;7(10501).
9. Zhou Q, Munger ME, Veenstra RG, Weigel BJ, Hirashima M, Munn DH, Murphy WJ, Azuma M, Anderson AC, Kuchroo VK, et al. Coexpression of Tim-3 and PD-1 identifies a CD8+ T-cell exhaustion phenotype in mice with disseminated acute myelogenous leukemia. *Blood.* 2011;117(17):4501-10.
10. Forsberg L, Leeb L, Thoren S, Morgenstern R, and Jakobsson P. Human glutathione dependent prostaglandin E synthase: gene structure and regulation. *FEBS Lett.* 2000;471(1):78-82.
11. Naraba H, Yokoyama C, Tago N, Murakami M, Kudo I, Fueki M, Oh-Ishi S, and Tanabe T. Transcriptional regulation of the membrane-associated prostaglandin E2 synthase gene. Essential role of the transcription factor Egr-1. *J Biol Chem.* 2002;277(32):28601-8.
12. Diaz-Munoz MD, Osma-Garcia IC, Cacheiro-Llaguno C, Fresno M, and Iniguez MA. Coordinated up-regulation of cyclooxygenase-2 and microsomal prostaglandin E synthase 1 transcription by nuclear factor kappa B and early growth response-1 in macrophages. *Cell Signal.* 2010;22(10):1427-36.
13. Furuya H, Wada M, Shimizu Y, Yamada PM, Hannun YA, Obeid LM, and Kawamori T. Effect of sphingosine kinase 1 inhibition on blood pressure. *FASEB J.* 2013;27(2):656-64.
14. Wang B, Chen J, Santiago FS, Janes M, Kavurma MM, Chong BH, Pimanda JE, and Khachigian LM. Phosphorylation and acetylation of histone H3 and autoregulation by early growth response 1 mediate interleukin 1beta induction of early growth response 1 transcription. *Arterioscler Thromb Vasc Biol.* 2010;30(3):536-45.
15. Ahmadzadeh M, Johnson LA, Heemskerk B, Wunderlich JR, Dudley ME, White DE, and Rosenberg SA. Tumor antigen-specific CD8 T cells infiltrating the tumor express high levels of PD-1 and are functionally impaired. *Blood.* 2009;114(8):1537-44.

16. Gros A, Robbins PF, Yao X, Li YF, Turcotte S, Tran E, Wunderlich JR, Mixon A, Farid S, Dudley ME, et al. PD-1 identifies the patient-specific CD8(+) tumor-reactive repertoire infiltrating human tumors. *J Clin Invest*. 2014;124(5):2246-59.
17. Paley MA, Kroy DC, Odorizzi PM, Johnnidis JB, Dolfi DV, Barnett BE, Bikoff EK, Robertson EJ, Lauer GM, Reiner SL, et al. Progenitor and terminal subsets of CD8+ T cells cooperate to contain chronic viral infection. *Science*. 2012;338(6111):1220-5.
18. Twyman-Saint Victor C, Rech AJ, Maity A, Rengan R, Pauken KE, Stelekati E, Benci JL, Xu B, Dada H, Odorizzi PM, et al. Radiation and dual checkpoint blockade activate non-redundant immune mechanisms in cancer. *Nature*. 2015;520(7547):373-7.
19. Sharma P, Hu-Lieskovan S, Wargo JA, and Ribas A. Primary, Adaptive, and Acquired Resistance to Cancer Immunotherapy. *Cell*. 2017;168(4):707-23.
20. Lin RL, and Zhao LJ. Mechanistic basis and clinical relevance of the role of transforming growth factor-beta in cancer. *Cancer Biol Med*. 2015;12(4):385-93.
21. Massague J. A very private TGF-beta receptor embrace. *Mol Cell*. 2008;29(2):149-50.
22. Hou W, Sampath P, Rojas JJ, and Thorne SH. Oncolytic Virus-Mediated Targeting of PGE2 in the Tumor Alters the Immune Status and Sensitizes Established and Resistant Tumors to Immunotherapy. *Cancer Cell*. 2016;30(1):108-19.
23. Zelenay S, van der Veen AG, Bottcher JP, Snelgrove KJ, Rogers N, Acton SE, Chakravarty P, Girotti MR, Marais R, Quezada SA, et al. Cyclooxygenase-Dependent Tumor Growth through Evasion of Immunity. *Cell*. 2015;162(6):1257-70.
24. Kim SH, Roszik J, Cho SN, Ogata D, Milton DR, Peng W, Menter D, Ekmekcioglu S, and Grimm EA. The COX2 effector microsomal PGE2 synthase-1 is a regulator of immunosuppression in cutaneous melanoma. *Clin Cancer Res*. 2018.
25. Fourcade J, Sun Z, Benallaoua M, Guillaume P, Luescher IF, Sander C, Kirkwood JM, Kuchroo V, and Zarour HM. Upregulation of Tim-3 and PD-1 expression is associated with tumor antigen-specific CD8+ T cell dysfunction in melanoma patients. *J Exp Med*. 2010;207(10):2175-86.
26. Sakuishi K, Apetoh L, Sullivan JM, Blazar BR, Kuchroo VK, and Anderson AC. Targeting Tim-3 and PD-1 pathways to reverse T cell exhaustion and restore anti-tumor immunity. *J Exp Med*. 2010;207(10):2187-94.
27. Hill JA, Feuerer M, Tash K, Haxhinasto S, Perez J, Melamed R, Mathis D, and Benoist C. Foxp3 transcription-factor-dependent and -independent regulation of the regulatory T cell transcriptional signature. *Immunity*. 2007;27(5):786-800.

Reviewers' comments:

Reviewer #1; cancer immunology:

Following the previous review, the investigators have provided more data to support their conclusion. Overall, the role of SK1 in modulating immune responses and tumor progression is very complex, and the mechanisms of action remain uncertain. The revised version and the new data of the manuscript raised many serious concerns.

MAJOR CONCERNS

1. The correlation of SK1 expression with survival with arbitrary cut off at 50% will need to be further evaluated with evaluation of all potential confounding variables including disease stage and tumor burden.
2. Many observations are truly striking and make this reviewer wonder about the relevance of the mouse tumor model in this study. These include:
 - a. The high-frequency Tregs in tumor transfected with sh controls (approximately 40% CD4+ T cells) in Figure 2G and Figure 4C.
 - b. The observation that only 10% CD8+ TILs express PD-1 in control Yumm tumors (Figure 3D) as one would expect that the majority of CD8+ TILs would be PD-1+.
 - c. The high spontaneous CD8+ TIL proliferation in control tumors (nearly 60% Ki67+ CD8+T cells) in Figure 3.
 - d. The high frequency of IFN-g producing CD8+ T cells without in vitro restimulation is also surprising. Also, these frequencies increased little upon SK1 deletion.
3. The evaluation of T cells in tumors at day 11 has evaluated small-size tumors as depicted in Figure 4A, which may render the frequencies assessment difficult. One would like to know the absolute number of cells.
4. The decreased of IR expression upon ICB is puzzling. The reasons supporting this observation remain elusive.
5. The mechanisms of action of SK1 deletion alone or in combination with ICB remains elusive. The data would suggest that SK1 deletion and CTLA-4 blockade may allow better Treg depletion to explain clinical observed. The mechanisms supporting the effect of SK1 removal together with PD-1 blockade remains uncertain.
6. In Figure suppl 4, Sk1 deletion appeared to correlate with increased IFN-g+ CD8+ T cells but decreased TNF-producing CD8+ TILs. This observation would suggest that Sk1 deletion would increase T cell dysfunction. These data do not fit well with the central hypothesis supported by the investigators.
7. The investigators have not included the evaluation of IL-2, which may be critical to evaluate T cell function and exhaustion.
8. The downregulation of PD-L1 expression by tumor cells in sh SK1 tumors together with to increased IFN-g producing CD8+ T cells is surprising and remains unexplained.

Reviewer #2; cancer immunology/melanoma:

In the revised version of the MS submitted by Caroline Imbert and colleagues have addressed

many of my concerns. They now provide detailed information on the patient cohort studied, the rescue-experiments are convincing as well as cell depletion experiments. Overall the quality of the MS has improved. However multiple concerns regarding the mechanism of action and the human relevance remain.

In the rebuttal letter the authors state:

"1. There are no differences in SPHK1 high and SPHK1 low group for tumor stage, histological subtype, and mutation status (BRAF and NRAS)."

I disagree with this statement. In the new Table 1 the authors show 81.9% of SK1 High patients are IVb/c compared to only 38.1% of SK1 low patients. This data suggests to me that SK1 is upregulated during disease progression (late stage) at least in a fraction of patients. Moreover, only 9.1 % and 27.3 % of SK1 high patients have a BRAF or NRAS mutation, respectively. Compared to 40 % and 43.8 % in SK1 low patient. This data suggests to me a possible correlation between SK1 expression and the absence of BRAF/NRAS pathway activity. The cohort of patients is small, thus the study might lack the power to address these interesting points, but I think the authors should at least discuss them in their MS as it indicates a possible regulation of SK1 in melanoma, which could be followed up in subsequent studies.

The authors also state:

"We performed additional experiments to address this very important point in order to analyze the expression of PD-1+ Eomes+ CD8 T cells that have been defined as an exhausted T cells in chronic infection and in melanoma (17, 18). Whereas we didn't observe Eomes+ CD8+ T cells at day 11, we observed a decreased frequency of PD-1+ Eomes+ and PD-1+ TIM3+ CD8 T cells at day 23 in shSK1 group that could reflect a decreased exhaustion phenotype in SK1 knockdown tumors at late time points (8, 9). Since this tenet remains to be confirmed by additional experiments, we prefer not to depict those data."

I think these data is very important for the MS. I would like to see these results repeated in another experiment. I would also like to see the day 11 data. These findings are in line with current thoughts that T cell exhaustion is a long-term process. If the data are reproducible, that in SK1 deficient tumors T cell exhaustion is impaired and effector function is maintained, this would provide more mechanistic insights.

In the figure provided in the rebuttal letter the authors show no correlation with PD-L1 expression nor with CD8 T cell infiltration. Why is this? Many other papers show a correlation at least between CD8 T cell infiltration and response to I/O. This also contradicts the pre-clinical findings, as the authors clearly see more CD8 T cells in SK1 low tumors. Can the authors please clarify this discrepancy? Along those lines in Figure 6 the authors focus on the correlation of SPHK1 with several immune suppressive marker. What would the correlation of SPHK1 with IFNg, GrzA, Perf, CD8 look like? One would assume it should be an anti-correlation.

"We apologize for this omission, CD8-deficient mice are CD8a Knockout mice and have been kindly given by Prof. J. van Meerwijk, INSERM U1043, Toulouse, France."

CD8-deficient C57BL/6 mice (a gift from Prof. J. van Meerwijk, INSERM U1043, Toulouse, France) Unfortunately, the authors did not write CD8a-deficient mice in the MM section of the MS. There is also no citation referring to the origin/phenotype of the mouse. They must cite the exact mouse strain they have used, as this is an important experimental detail. Have the authors used the Tak Mak mouse? More importantly, the CD8a-KO mouse has also impaired cross presenting DCs, thus this model is not ideally suited. The authors should move the CD8a-ko mouse graph into the supplementary figure and bring the experiments using antibody-mediated depletion into the main figure.

In line with this, there are no information's provided on the culture conditions of the various cell lines used in the study. In the section Murine cell lines, MC38 and 4T1.2 do not even appear.

Figure 3 suggests, that SK1 downregulation affects the recruitment and phenotype of Tregs and thus the effector function of CD8+ T cells. This data still remains descriptive and correlative. The authors should inject ctrl and SK1 cells into Foxp3-DTR mice to show the effect of SK1 knockdown

if no Tregs are infiltrating tumours. This would, again strengthen the MS significantly. In line with finding, I believe the correlation of SK1 expression and the presence of Foxp3+ Tregs in human melanomas is important to show. Which antibody did the authors use? The clone PCH101, and Ab20034 as well as Ab10563 have been shown to work reliable. The authors should aim to substantiate the notion that SK1 through currently unknown mechanisms influences Treg recruitment in preclinical and human melanoma. Thus, leading to improved anti-tumour immunity.

“In accordance with the results obtained with the knockdown strategy (Fig. 4A), the pharmacological approach showed that SK1 inhibition sensitizes melanoma tumors more efficiently to anti-CTLA-4 than anti-PD-1 treatment (Fig. 4G)”

Unfortunately, there is no panel G in Figure 4. The authors probably mean Figure 4F.

In figure 4 the colour code of the box plots in panel C – E are missing. Based on the colour code/legend in B I could assess the figures, but it would be appropriate to depict the legend at least once in this figure. The authors also didn't provide any data on how SK1i treatment affected Treg recruitment, Chemokine production and CD8 effector phenotype. So far the authors have not provided convincing data that there SK1 inhibitor has a similar mode of action compared to genetic silencing. Thus, there data does not support their conclusion.

The authors would need to show TIL analyses after SK1i treatment showing reduced Treg infiltration, lower levels of CCL17 and CCL22, more PD1+CTLA4+Tim3+ CD8 T cells etc. This data set is important to address the possible translational relevance of their findings.

“We agree with the Reviewer that it is necessary to demonstrate that the inhibitory effect of the shRNA against Sphk1 on the expression of Ptges is not an off-target effect. Establishment of a full SK1 KO Yumm cell line, using the CRISPR/Cas9 technology, is not possible as the deletion of SK1 highly alters cell proliferation.”

What do the authors mean by this statement? How does a genetic knockout of Sphk1 influence cell proliferation? Increase? Decrease? Is there something special about the Yumm cell line?

In response to reviewer 1 point 3 the authors provided their gating strategy for the assessment of Ki67 in T cells. I am very surprised to see that 73.3% of all viable cells in their tumours are CD45+ immune cells? This can't be right. What is the fraction of viable cells? Based on the provided dot plots, it looks like the authors have a significant amount of dead cells in their analyses. Can the author please clarify, why they have so many dead cells in their preparations and why the 75% of cells are immune and only 25% are tumour cells?

In Figure 6 again the appropriate labelling of panel A the fold-change scale is missing.

Response to Reviewers' Comments:

The point-by-point response to reviewers' comments is highlighted in blue and bold.

Reviewer #1; cancer immunology:

Following the previous review, the investigators have provided more data to support their conclusion, Overall, the role of SK1 in modulating immune responses and tumor progression is very complex, and the mechanisms of action remain uncertain. The revised version and the new data of the manuscript raised many serious concerns.

MAJOR CONCERNS

1. The correlation of SK1 expression with survival with arbitrary cut off at 50% will need to be further evaluated with evaluation of all potential confounding variables including disease stage and tumor burden

As requested by Reviewer #1 during the first round of revision, the major variables including disease stage, BRAF or NRAS mutation, Breslow and ulceration status as well as available treatment information from the study cohort were presented in Table 1. This table has now been amended by introducing the chi-squared or Fisher's exact test used to compare categorical variables. However, considering the size of our cohort, multivariate analysis including all potential confounding variables cannot be performed. This would indeed violate the 10 events per 1 variable rule (Peduzzi, Concato et al. 1995), inducing greater risks of bias. Importantly, we are currently conducting a prospective clinical trial (IMMUSPHINX: NCT03627026), which will allow us to extend our observations to additional tumor specimens from metastatic melanoma patients treated with immune checkpoint inhibitors. This point is now mentioned in the discussion (page 11).

2. Many observations are truly striking and make this reviewer wonder about the relevance of the mouse tumor model in this study. These include:
a. The high-frequency Tregs in tumor transfected with sh controls (approximately 40% CD4+ T cells) in Figure 2G and Figure 4C.

Mouse melanoma models are well described for their propensity to promote regulatory responses. As a matter of fact, similar tumor Treg frequencies were reported using other mouse melanoma (e.g., B16-F10 or B16-OVA) models (Klages, Mayer et al. 2010, Oh, Kim et al. 2017). In addition, Yumm 1.7 melanoma cells carry the BRAF^{V600E} mutation. This mutation has already been reported to promote infiltration of tumors by Treg in spontaneous mouse melanoma models (Shabaneh, Molodtsov et al. 2018). Furthermore, in patients with metastatic melanoma up to 70% of CD4+ TILs were shown to express Foxp3 (Ahmadzadeh, Felipe-Silva et al. 2008).

b. The observation that only 10% CD8+ TILs express PD-1 in control Yumm tumors (Figure 3D) as one would expect that the majority of CD8+ TILs would be PD-1+.

As shown in Figure 3D, the frequency of CD8+ PD-1+ T cells in control Yumm tumors is less than 20% because TIL content was analysed on day 11 after

tumor cell injection. As expected, we obviously observed much higher frequencies at day 20 (see figure A below).

Figure A (not to be included in the manuscript). Analysis of PD-1 expression in CD8 TILs at day 11 (D11) and day 20 (D20) after inoculation of shCtrl Yumm cells.

c. The high spontaneous CD8+ TIL proliferation in control tumors (nearly 60% Ki67+ CD8+ T cells) in Figure 3.

This point was already discussed in the first revised version of the manuscript. We also previously explained the gating strategy including isotype control staining for Ki67 expression in T cells isolated from tumors and draining lymph nodes (TDLN). The high frequency of Ki67+ CD8+ T cells observed at day 11 corresponds to the early phase of T cell activation. Similar percentages of Ki67+ CD8+ TILs were observed in the well-validated B16F10 mouse melanoma model, by us (see Figure B below) as well as by others (Zamarin, Ricca et al. 2018).

Figure B (not to be included in the manuscript). Analysis of Ki67 expression in CD8+ TILs at day 11 after inoculation of shCtrl Yumm or B16F10 cells.

d. The high frequency of IFN-g producing CD8+ T cells without *in vitro* restimulation is also surprising. Also, these frequencies increased little upon SK1 deletion.

Actually, IFN- γ production was analysed after *in vitro* restimulation with PMA and ionomycin as described in the Methods section (page 17) and detailed again in the Results section (page 7). To avoid any confusion, this technical point has been added in the legends to Figure 3 and Supplementary Figure 5. This polyclonal and strong stimulation can therefore explain the high frequency of IFN- γ + CD8+ T cells. We agree with Reviewer #1 that these frequencies slightly, yet significantly, increased upon SK1 silencing.

3. The evaluation of T cells in tumors at day 11 has evaluated small-size tumors as

depicted in Figure 4A, which may render the frequencies assessment difficult. One would like to know the absolute number of cells.

Tumor weights at day 11 averaged 100 mg (irrespectively of SK1 silencing). Importantly, we selected this particular time point to assess the tumor immune infiltrate just before tumors evade from the immune system. As requested by the Reviewer, we calculated the absolute number of T cells at day 11 and found no differences between the groups, expected for the Treg cell absolute number that was dramatically decreased in the shSK1 tumor group.

Figure C (not to be included in the manuscript). Analysis of absolute numbers at day 11 after inoculation of either shCtrl or shSK1 Yumm cells. The absolute number of cells per mg of tumor sample was determined using CountBright™ Absolute Counting Beads (Life technologies). The following formula was used for this calculation: absolute cell number = number of cells acquired/(tumor wet weight/[number of beads acquired/number of beads added to sample]) (Simpson, Li et al. 2013).

4. The decreased of IR expression upon ICB is puzzling. The reasons supporting this observation remain elusive.

Indeed, we have no clear explanation for this observation. It is unlikely due to a competition of binding to PD-1 or CTLA-4 between the therapeutic antibodies and the antibodies used for staining since they recognize different epitopes. As these data (ex Figure 4E) might be confusing and do not clarify the mechanisms by which SK1 knockdown enhances ICI therapy, we decided to delete them from the manuscript.

5. The mechanisms of action of SK1 deletion alone or in combination with ICB remains elusive. The data would suggest that SK1 deletion and CTLA-4 blockade may allow better Treg depletion to explain clinical observed. The mechanisms supporting the effect of SK1 removal together with PD-1 blockade remains uncertain.

Perhaps, as presented in the previous version of Figure 4C, the effects of SK1 silencing on the Treg depletion by anti-PD-1 were not well appreciated by the Reviewer. These effects can be clearly seen when presented in a semi-logarithmic scale (see below, new presentation of Figure 4C). As compared to ICI alone or SK1 knockdown alone, combined therapies led to a significant additive reduction of Treg depletion. We thus propose to replace the previous Figure 4C by this new type of presentation.

Figure 4C. Percent of total CD4+Foxp3+ cells. Data are from Figure 4C middle panel, presented using a semi-logarithmic scale to emphasize the additive effect of SK1 silencing on the response to ICI.

As a matter of fact, Treg recruitment into the tumor microenvironment was reported to impede both the antitumor immune response and the response to anti-PD-1 therapy. Therefore, clinical trials aiming at limiting Treg accumulation, using for instance, anti-CCR4 antibodies either as monotherapy or in combination with ICI, are being conducted (Sugiyama, Nishikawa et al. 2013, Ueda 2015).

In order to confirm that Treg play an important role in our model, we orthotopically grafted shCtrl and shSK1 Yumm cells in DEREK (Foxp3-DTR-GFP) mice. Following diphtheria toxin (DT) injection, a highly efficient depletion of Treg was obtained both in the periphery and tumors at day 8 (see new Supplementary Fig. 4; below). This led to total rejection of the tumors in shCtrl and shSK1 groups.

A

B

C

Supplementary Figure 4. Impact of Foxp3+ Treg depletion on Yumm tumor growth. shCtrl or shSK1(1) Yumm cells were intradermally injected in DEREK or WT mice on day 0: (A) Mice were i.p. injected with 1µg of diphtheria toxin (DT) or PBS on days 1, 4, 7 and 10. (B) Analysis of Foxp3+ CD4+ T cells at day 8 in the tumor (TUM) and tumor draining lymph nodes (TDLN) of DEREK mice injected with PBS or DT at

days 1, 4 and 7. (C) Tumor growth of DEREK or WT mice (n=6-12). Tumor volumes are presented as means \pm SEM. Samples were compared using a two-way ANOVA test.

These novel results clearly demonstrate that Treg are a major immunosuppressive population that impairs anti-tumor immune responses in this model. Altogether, our results support the view that SK1 silencing enhances ICI response via a Treg-mediated mechanism.

This is further corroborated by our finding that SK1 silencing down-regulates PTGES expression and subsequently attenuates PGE2 production (Figure 5). As the PGE2 prostaglandin is known to facilitate Treg accumulation in tumors (refs), this further establishes a mechanistic link between SK1 silencing and Treg reduction. Moreover, we show that SK1 silencing was associated with a reduced production of immunosuppressive chemokines CCL17 and CCL22 (Figure 4D), which potently promote Treg infiltration (Curiel, Coukos et al. 2004, Ishida and Ueda 2006).

6. In Figure suppl 4, Sk1 deletion appeared to correlate with increased IFN-g+ CD8+ T cells but decreased TNF-producing CD8+ TILs. This observation would suggest that Sk1 deletion would increase T cell dysfunction. These data do not fit well with the central hypothesis supported by the investigators.

Our central hypothesis is that a high tumor SK1 expression induces an immunosuppressive microenvironment and our main conclusion is that “targeting SK1 decreases Treg accumulation and enhances ICI efficacy in mouse melanoma”.

Whereas reducing the TNF production may be considered as the result of a T cell dysfunction, we previously demonstrated that TNF promotes the tumor growth of melanoma cell lines, including the Yumm melanoma cells, by limiting the accumulation of CD8+ TILs (Bertrand, Rochotte et al. 2015). Moreover, we showed that TNF triggers the resistance to anti-PD-1 therapy and TNF blockade overcomes this resistance in mouse cancer models (Bertrand, Montfort et al. 2017). This concept has been recently confirmed by Melero and co-workers (Perez-Ruiz, Minute et al. 2019).

In this study, our goal was not to dissect the role SK1 from melanoma cells may play in T cell exhaustion (dysfunction). Analysis of such a phenomenon would need an evaluation at later stages than examined here and in a situation of adaptive resistance. Determining the precise interconnection(s) linking tumor-dependent SK1 expression and the modulation of anti-tumor T cells effector functions will be the subject of separate study.

7. The investigators have not included the evaluation of IL-2, which may be critical to evaluate T cell function and exhaustion.

We evaluated the frequencies of IL-2 positive T cells but we found very few IL-2+ CD8 TILs (around 2% of CD8+ T cells) and no differences between shCtrl and shSK1 groups (data not shown).

8. The downregulation of PD-L1 expression by tumor cells in sh SK1 tumors together with to increased IFN- γ producing CD8+ T cells is surprising and remains unexplained.

We agree with Reviewer #1 that IFN- γ is known to induce PD-L1 expression in tumor cells. Under our experimental conditions, whereas we found an increased IFN- γ production by CD8+ TILs from shSK1 tumors following PMA-ionomycin stimulation of TILs, we noticed a significant reduction of TNF, a cytokine also known to promote PD-L1 expression in melanoma cells (Lim, Li et al. 2016). However, there are known additional regulators of PD-L1 expression such as HIF1 α , Stat3 and NF- κ B (Wang, Wang et al. 2018) that can be activated by SK1-dependent signalling pathways (Ader, Brizuela et al. 2008, Alvarez, Harikumar et al. 2010, Liang, Nagahashi et al. 2013). We agree with Reviewer #1 this is a very interesting observation, however out of the scope of the present study.

Reviewer #2; cancer immunology/melanoma:

In the revised version of the MS submitted by Caroline Imbert and colleagues have addressed many of my concerns. They now provide detailed information on the patient cohort studied, the rescue-experiments are convincing as well as cell depletion experiments. Overall the quality of the MS has improved. However multiple concerns regarding the mechanism of action and the human relevance remain.

In the rebuttal letter the authors state:

“1. There are no differences in SPHK1 high and SPHK1 low group for tumor stage, histological subtype, and mutation status (BRAF and NRAS).” I disagree with this statement. In the new Table 1 the authors show 81.9% of SK1 High patients are IVb/c compared to only 38.1% of SK1 low patients. This data suggests to me that SK1 is upregulated during disease progression (late stage) at least in a fraction of patients. Moreover, only 9.1 % and 27.3 % of SK1 high patients have a BRAF or NRAS mutation, respectively. Compared to 40 % and 43.8 % in SK1 low patient. This data suggests to me a possible correlation between SK1 expression and the absence of BRAF/NRAS pathway activity. The cohort of patients is small, thus the study might lack the power to address these interesting points, but I think the authors should at least discuss them in their MS as it indicates a possible regulation of SK1 in melanoma, which could be followed up in subsequent studies.

We agree with Reviewer #2 that evaluating whether SK1 expression in tumors is related to the patient BRAF and NRAS mutational status is of particular importance. However, our cohort does not allow for such an analysis. First, in this cohort most tumors are not mutated for BRAF (22 out of 31). Second, among the 9 patients having a BRAF mutation, only one is associated with the “SK1 high” group, thus precluding a statistically relevant comparison. A much larger cohort would be necessary to address this issue. As mentioned in the response to Reviewer #1 (item 1), we are currently conducting a prospective clinical trial (IMMUSPHINX: NCT03627026), which will allow us (i) to confirm the

prognostic significance of SK1 and (ii) to link this parameter to other parameters such as treatment, stage and mutational status.

Similarly, a misinterpretation can be made when comparing the expression of SK1 in patients with different metastatic subclasses. For instance, for stage IVb there is a single patient in the « SK1 low » group. In contrast, when both groups contain a reasonable number of patients such as for stage IVc (12 patients), one could conclude that there is no difference between the two groups.

As recommended by Reviewer #2, the following sentence has been introduced in the Discussion (page 11): “Considering the small number of patients, the association between the expression of SK1 and the tumor stage, the mutation status of BRAF and NRAS as well as immune responses could not be performed due to weak statistical power in this retrospective study. This will be performed on a larger cohort of advanced melanoma patients treated with anti-PD-1 in combination or not with anti-CTLA-4, from a prospective clinical trial (IMMUSPHINX: NCT03627026) we are currently conducting in our institute”.

The authors also state:

“We performed additional experiments to address this very important point in order to analyze the expression of PD-1+ Eomes+ CD8 T cells that have been defined as an exhausted T cells in chronic infection and in melanoma (17, 18). Whereas we didn't observe Eomes+ CD8+ T cells at day 11, we observed a decreased frequency of PD-1+ Eomes+ and PD-1+ TIM3+ CD8 T cells at day 23 in shSK1 group that could reflect a decreased exhaustion phenotype in SK1 knockdown tumors at late time points (8, 9). Since this tenet remains to be confirmed by additional experiments, we prefer not to depict those data.”

I think these data is very important for the MS. I would like to see these results repeated in another experiment. I would also like to see the day 11 data. These findings are in line with current thoughts that T cell exhaustion is a long-term process. If the data are reproducible, that in SK1 deficient tumors T cell exhaustion is impaired and effector function is maintained, this would provide more mechanistic insights.

As requested by Reviewer #2, we performed additional experiments to analyse the frequencies of PD-1+ Eomes+ CD8+ T cells (Figure D below). We observed few PD-1+ Eomes+ CD8+ T cells at day 11 irrespective of the condition; however these frequencies significantly increased by day 20. In this experiment, we did not find any differences in the frequencies of PD-1+ Eomes+ CD8+ T cells between shCtrl and shSK1 tumors.

Figure D (not to be included in the manuscript). Flow cytometry analysis of PD1+ Eomes + CD8 T cells at day 11 and day 20 after inoculation of either shCtrl or shSK1 Yumm cells.

Thus, this T cell exhaustion hypothesis cannot explain the regression of tumors observed at day 11 upon SK1 silencing and ICI treatment. Reversing exhaustion of T cells (a long-term process) does not seem to be the main mechanism through which SK1 inhibition promotes ICI efficacy at early stages, as opposed to the dramatic impact SK1 silencing has on the Treg population.

In the figure provided in the rebuttal letter the authors show no correlation with PD-L1 expression nor with CD8 T cell infiltration. Why is this? Many other papers show a correlation at least between CD8 T cell infiltration and response to I/O. This also contradicts the pre-clinical findings, as the authors clearly see more CD8 T cells in SK1 low tumors. Can the authors please clarify this discrepancy?

We agree with Reviewer #2 that CD8 T cell infiltration could be associated with a good prognosis, as described by Tumeh et al. (Tumeh, Harview et al. 2014), who have used a cohort in which patients are clearly defined as responders or non-responders. By using the recommended classification of TILs (brisk, non-brisk; (Mihm and Mule 2015)), no correlation between CD8+ cells and survival was seen. Nonetheless, PD-L1 expression in human tumors tended to correlate with progression-free survival (see Figure E below). By extending the cohort, our prospective clinical trial Immusphinx (NCT03627026) will help investigate these relationships.

Figure E (not to be included in the manuscript). Progression-free survival curve of patients with more than 1% of melanoma cells positive for PD-L1 (black line; n=15) or less than 1% (red line; n=14). Survival times were calculated from the first day of the cycle of anti-PD-1 post biopsy.

Along those lines in Figure 6 the authors focus on the correlation of SPHK1 with several immune suppressive marker. What would the correlation of SPHK1 with IFNg, GrzA, Perf, CD8 look like? One would assume it should be an anti-correlation.

This anti-correlation can be expected but not always seen in human tumors. As a matter of fact, it is well established that expression of immune regulatory molecules in the tumor microenvironment is positively associated with infiltration of tumors with immune cells such as T cells across multiple cancers (Trujillo, Sweis et al. 2018). We did not find an anti-correlation for IFNg, GrzA, Perf, CD8 but instead a weak correlation (Table A below). These findings are not surprising since TILs including CD8 and as well as Treg are enriched in hot (inflamed) tumors such as melanoma, a phenomenon Gajewski’s group demonstrated when showing a remarkable positive correlation between Foxp3 expression and CD8+ cell infiltration in tumors (Spranger, Spaapen et al. 2013).

SPHK1	IFNG	GZMA	PRF1	CD8A	CD8B
r	0.1679	0.2073	0.2264	0.1744	0.1992
p value	0.0016	0.0001	0.0000	0.0012	0.0002

Table A. (not to be included in the manuscript). Correlations between SPHK1 and IFNG, GZMA, PRF1, CD8A and CD8 gene expression in tumors from metastatic melanoma patients using Spearman’s rank-order correlation (n=342).

“We apologize for this omission, CD8-deficient mice are CD8 α Knockout mice and have been kindly given by Prof. J. van Meerwijk, INSERM U1043, Toulouse, France.”

CD8-deficient C57BL/6 mice (a gift from Prof. J. van Meerwijk, INSERM U1043, Toulouse, France)

Unfortunately, the authors did not write CD8a-deficient mice in the MM section of the MS. There is also no citation referring to the origin/phenotype of the mouse. They must cite the exact mouse strain they have used, as this is an important experimental detail. Have the authors used the Tak Mak mouse? More importantly, the CD8a-KO

mouse has also impaired cross presenting DCs, thus this model is not ideally suited. The authors should move the CD8a-ko mouse graph into the supplementary figure and bring the experiments using antibody-mediated depletion into the main figure.

As requested by Reviewer #2, we deleted the data with the CD8 α deficient mice as the CD8 depleting experiments using an antibody are more convincing and give similar results (Supplementary Figure 1B).

In line with this, there are no information's provided on the culture conditions of the various cell lines used in the study. In the section Murine cell lines, MC38 and 4T1.2 do not even appear.

MC38 and 4T1 were maintained in DMEM supplemented with 10% FCS and L-glutamine. This information has been added in the supplementary methods.

Figure 3 suggests, that SK1 downregulation affects the recruitment and phenotype of Tregs and thus the effector function of CD8+ T cells. This data still remains descriptive and correlative. The authors should inject ctrl and SK1 cells into Foxp3-DTR mice to show the effect of SK1 knockdown if no Tregs are infiltrating tumours. This would, again strengthen the MS significantly. In line with finding, I believe the correlation of SK1 expression and the presence of Foxp3+ Tregs in human melanomas is important to show. Which antibody did the authors use? The clone PCH101, and Ab20034 as well as Ab10563 have been shown to work reliable. The authors should aim to substantiate the notion that SK1 through currently unknown mechanisms influences Treg recruitment in preclinical and human melanoma. Thus, leading to improved anti-tumour immunity.

Considering the large proportion Treg represent among the total pool of TILs in mice, it is highly conceivable that a reduction in their proportion favors response to ICI in our model. In order to prove that Treg play an important role in this model, we orthotopically grafted shCtrl and shSK1 Yumm cells in DEREK (Foxp3-DTR-GFP) mice. Following diphtheria toxin (DT) injection, a highly efficient depletion of Treg in periphery and tumors was observed, which resulted in tumor rejection in both groups of animals. These new data, presented in Supplementary Figure 4 (see also response to Reviewer #1, point 5), now clearly establish that Treg lymphocytes act as a major immunosuppressive population in our experimental preclinical model. Given our previous results demonstrating the effect of SK1 knockdown in combination either with anti-CTLA-4 or anti-PD-1 on the reduction of Treg infiltration (Figure 4C), these new findings support the view that SK1 silencing enhances ICI response via a Treg-mediated mechanism.

Regarding human melanoma samples, Foxp3 staining was performed with the 236A/E7 clone as previously described (Martinet, Le Guellec et al. 2012) (see Table B below). A significant ($p = 0.037$) association between SPHK1 expression and Foxp3 staining was observed. Although these data are interesting, a much larger cohort (such as the prospective clinical trial

Immusphinx NCT03627026) would be necessary to address this issue. We thus would prefer not to include these data in the body of the manuscript.

	Total N=32	Low SPHK1 (<=50%) N= 21	High SPHK1 (>50%) N= 11
Foxp3 score (n=32)			p = 0.0367
negative or weak	27 (84.4%)	20 (95.2%)	7 (63.6%)
positive (>1%)	5 (15.6%)	1 (4.8%)	4 (36.4%)

Table B. (not to be included in the manuscript). Foxp3 expression in tumor samples exhibiting low and high SPHK1 expression.

“In accordance with the results obtained with the knockdown strategy (Fig. 4A), the pharmacological approach showed that SK1 inhibition sensitizes melanoma tumors more efficiently to anti-CTLA-4 than anti-PD-1 treatment (Fig. 4G)” Unfortunately, there is no panel G in Figure 4. The authors probably mean Figure 4F.

This panel is now panel E in the new Figure 4.

In figure 4 the colour code of the box plots in panel C – E are missing. Based on the colour code/legend in B I could assess the figures, but it would be appropriate to depict the legend at least once in this figure.

As requested by Reviewer #2, we added the colour code in the new Figure 4.

The authors also didn’t provide any data on how SK1i treatment affected Treg recruitment, Chemokine production and CD8 effector phenotype. So far the authors have not provided convincing data that there SK1 inhibitor has a similar mode of action compared to genetic silencing. Thus, there data does not support their conclusion.

The authors would need to show TIL analyses after SK1i treatment showing reduced Treg infiltration, lower levels of CCL17 and CCL22, more PD1+CTLA4+Tim3+ CD8 T cells etc. This data set is important to address the possible translational relevance of their findings.

As requested by Reviewer #2, we analysed TILs at day 11 after SKI-I and anti-CTLA-4 treatment. We demonstrated that this combination resulted in a significantly higher CD8/Treg ratio in tumors (see Figure below). The latter results have been added in the new Figure 4F.

Figure 4F. Mice were challenged with untransfected Yumm cells on day 0, and then treated or not with vehicle, SKI-1, anti-CTLA-4. (F) CD8/CD4+ Foxp3+ ratio at day 11. Samples were compared using Kruskal-Wallis test with Dunn's correction.

“We agree with the Reviewer that it is necessary to demonstrate that the inhibitory effect of the shRNA against Sphk1 on the expression of Ptges is not an off-target effect. Establishment of a full SK1 KO Yumm cell line, using the CRISPR/Cas9 technology, is not possible as the deletion of SK1 highly alters cell proliferation.” What do the authors mean by this statement? How does a genetic knockout of Sphk1 influence cell proliferation? Increase? Decrease? Is there something special about the Yumm cell line?

SK1 expression/activity is tightly regulated by growth factors and favors cell proliferation (Pulkoski-Gross and Obeid 2018). Yumm cells exhibit a lower in vitro cell growth rate as compared to murine B16 melanoma cell lines. In this context, downregulation of SK1 in Yumm cells does not affect their proliferation as shown in Fig. 2A. Despite numerous efforts we failed to generate SK1 knockout Yumm cell lines by using the CrispR/Cas9 technology. In addition, to our knowledge no melanoma cell model has yet been published where SK1 has been completely deleted by CrispR/Cas9.

In response to reviewer 1 point 3 the authors provided their gating strategy for the assessment of Ki67 in T cells. I am very surprised to see that 73.3% of all viable cells in their tumours are CD45+ immune cells? This can't be right. What is the fraction of viable cells? Based on the provided dot plots, it looks like the authors have a significant amount of dead cells in their analyses. Can the author please clarify, why they have so many dead cells in their preparations and why the 75% of cells are immune and only 25% are tumour cells?

We apologize if the data were confusing. This is dependent on the gating strategy because in this panel we focused on TILs and the parameters (SSC-A, FSC-A and threshold) were adjusted for TIL analysis. Of course, there is not 73.3% of CD45+ viable cells in tumors. The percentage of CD45 positive cells is 10% of total cells as shown in the Figure F below. The percentage of dead cells is related to tumor necrosis and tumor digestion protocol (mechanic and enzymatic digestion).

Figure F (not to be included in the manuscript). Percentage of total CD45+ cells in shCtrl and shSK1 tumors at day 11.

In Figure 6 again the appropriate labelling of panel A the fold-change scale is missing.

The fold change has been added in the new Figure 6.

References

- Ader, I., L. Brizuela, P. Bouquerel, B. Malavaud and O. Cu villier (2008). "Sphingosine kinase 1: a new modulator of hypoxia inducible factor 1alpha during hypoxia in human cancer cells." *Cancer Res* **68**(20): 8635-8642.
- Ahmadzadeh, M., A. Felipe-Silva, B. Heemskerck, D. J. Powell, Jr., J. R. Wunderlich, M. J. Merino and S. A. Rosenberg (2008). "FOXP3 expression accurately defines the population of intratumoral regulatory T cells that selectively accumulate in metastatic melanoma lesions." *Blood* **112**(13): 4953-4960.
- Alvarez, S. E., K. B. Harikumar, N. C. Hait, J. Allegood, G. M. Strub, E. Y. Kim, M. Maceyka, H. Jiang, C. Luo, T. Kordula, S. Milstien and S. Spiegel (2010). "Sphingosine-1-phosphate is a missing cofactor for the E3 ubiquitin ligase TRAF2." *Nature* **465**(7301): 1084-1088.
- Bertrand, F., A. Montfort, E. Marcheteau, C. Imbert, J. Gilhodes, T. Filleron, P. Rochaix, N. Andrieu-Abadie, T. Levade, N. Meyer, C. Colacios and B. Segui (2017). "TNFalpha blockade overcomes resistance to anti-PD-1 in experimental melanoma." *Nat Commun* **8**(1): 2256.
- Bertrand, F., J. Rochotte, C. Colacios, A. Montfort, A. F. Tilkin-Mariame, C. Touriol, P. Rochaix, I. Lajoie-Mazenc, N. Andrieu-Abadie, T. Levade, H. Benoist and B. Segui (2015). "Blocking Tumor Necrosis Factor alpha Enhances CD8 T-cell-Dependent Immunity in Experimental Melanoma." *Cancer Res* **75**(13): 2619-2628.

Curiel, T. J., G. Coukos, L. Zou, X. Alvarez, P. Cheng, P. Mottram, M. Evdemon-Hogan, J. R. Conejo-Garcia, L. Zhang, M. Burow, Y. Zhu, S. Wei, I. Kryczek, B. Daniel, A. Gordon, L. Myers, A. Lackner, M. L. Disis, K. L. Knutson, L. Chen and W. Zou (2004). "Specific recruitment of regulatory T cells in ovarian carcinoma fosters immune privilege and predicts reduced survival." *Nat Med* **10**(9): 942-949.

Ishida, T. and R. Ueda (2006). "CCR4 as a novel molecular target for immunotherapy of cancer." *Cancer Sci* **97**(11): 1139-1146.

Klages, K., C. T. Mayer, K. Lahl, C. Loddenkemper, M. W. Teng, S. F. Ngiow, M. J. Smyth, A. Hamann, J. Huehn and T. Sparwasser (2010). "Selective depletion of Foxp3+ regulatory T cells improves effective therapeutic vaccination against established melanoma." *Cancer Res* **70**(20): 7788-7799.

Liang, J., M. Nagahashi, E. Y. Kim, K. B. Harikumar, A. Yamada, W. C. Huang, N. C. Hait, J. C. Allegood, M. M. Price, D. Avni, K. Takabe, T. Kordula, S. Milstien and S. Spiegel (2013). "Sphingosine-1-phosphate links persistent STAT3 activation, chronic intestinal inflammation, and development of colitis-associated cancer." *Cancer Cell* **23**(1): 107-120.

Lim, S. O., C. W. Li, W. Xia, J. H. Cha, L. C. Chan, Y. Wu, S. S. Chang, W. C. Lin, J. M. Hsu, Y. H. Hsu, T. Kim, W. C. Chang, J. L. Hsu, H. Yamaguchi, Q. Ding, Y. Wang, Y. Yang, C. H. Chen, A. A. Sahin, D. Yu, G. N. Hortobagyi and M. C. Hung (2016). "Deubiquitination and Stabilization of PD-L1 by CSN5." *Cancer Cell* **30**(6): 925-939.

Martinet, L., S. Le Guellec, T. Filleron, L. Lamant, N. Meyer, P. Rochaix, I. Garrido and J. P. Girard (2012). "High endothelial venules (HEVs) in human melanoma lesions: Major gateways for tumor-infiltrating lymphocytes." *Oncoimmunology* **1**(6): 829-839.

Mihm, M. C., Jr. and J. J. Mule (2015). "Reflections on the Histopathology of Tumor-Infiltrating Lymphocytes in Melanoma and the Host Immune Response." *Cancer Immunol Res* **3**(8): 827-835.

Oh, D. S., H. Kim, J. E. Oh, H. E. Jung, Y. S. Lee, J. H. Park and H. K. Lee (2017). "Intratumoral depletion of regulatory T cells using CD25-targeted photodynamic therapy in a mouse melanoma model induces antitumoral immune responses." *Oncotarget* **8**(29): 47440-47453.

Peduzzi, P., J. Concato, A. R. Feinstein and T. R. Holford (1995). "Importance of events per independent variable in proportional hazards regression analysis. II. Accuracy and precision of regression estimates." *J Clin Epidemiol* **48**(12): 1503-1510.

Perez-Ruiz, E., L. Minute, I. Otano, M. Alvarez, M. C. Ochoa, V. Belsue, C. de Andrea, M. E. Rodriguez-Ruiz, J. L. Perez-Gracia, I. Marquez-Rodas, C. Llacer, M. Alvarez, V. de Luque, C. Molina, A. Teijeira, P. Berraondo and I. Melero (2019). "Prophylactic TNF blockade uncouples efficacy and toxicity in dual CTLA-4 and PD-1 immunotherapy." *Nature* **569**(7756): 428-432.

Pulkoski-Gross, M. J. and L. M. Obeid (2018). "Molecular mechanisms of regulation of sphingosine kinase 1." *Biochim Biophys Acta Mol Cell Biol Lipids* **1863**(11): 1413-1422.

Shabaneh, T. B., A. K. Molodtsov, S. M. Steinberg, P. Zhang, G. M. Torres, G. A. Mohamed, A. Boni, T. J. Curiel, C. V. Angeles and M. J. Turk (2018). "Oncogenic BRAF(V600E) Governs Regulatory T-cell Recruitment during Melanoma Tumorigenesis." *Cancer Res* **78**(17): 5038-5049.

Simpson, T. R., F. Li, W. Montalvo-Ortiz, M. A. Sepulveda, K. Bergerhoff, F. Arce, C. Roddie, J. Y. Henry, H. Yagita, J. D. Wolchok, K. S. Peggs, J. V. Ravetch, J. P. Allison and S. A. Quezada (2013). "Fc-dependent depletion of tumor-infiltrating regulatory T cells co-defines the efficacy of anti-CTLA-4 therapy against melanoma." *J Exp Med* **210**(9): 1695-1710.

Spranger, S., R. M. Spaapen, Y. Zha, J. Williams, Y. Meng, T. T. Ha and T. F. Gajewski (2013). "Up-regulation of PD-L1, IDO, and T(regs) in the melanoma tumor microenvironment is driven by CD8(+) T cells." *Sci Transl Med* **5**(200): 200ra116.

Sugiyama, D., H. Nishikawa, Y. Maeda, M. Nishioka, A. Tanemura, I. Katayama, S. Ezo, Y. Kanakura, E. Sato, Y. Fukumori, J. Karbach, E. Jager and S. Sakaguchi (2013). "Anti-CCR4 mAb selectively depletes effector-type FoxP3+CD4+ regulatory T cells, evoking antitumor immune responses in humans." *Proc Natl Acad Sci U S A* **110**(44): 17945-17950.

Trujillo, J. A., R. F. Sweis, R. Bao and J. J. Luke (2018). "T Cell-Inflamed versus Non-T Cell-Inflamed Tumors: A Conceptual Framework for Cancer Immunotherapy Drug Development and Combination Therapy Selection." *Cancer Immunol Res* **6**(9): 990-1000.

Tumeh, P. C., C. L. Harview, J. H. Yearley, I. P. Shintaku, E. J. Taylor, L. Robert, B. Chmielowski, M. Spasic, G. Henry, V. Ciobanu, A. N. West, M. Carmona, C. Kivork, E. Seja, G. Cherry, A. J. Gutierrez, T. R. Grogan, C. Mateus, G. Tomasic, J. A. Glaspy, R. O. Emerson, H. Robins, R. H. Pierce, D. A. Elashoff, C. Robert and A. Ribas (2014). "PD-1 blockade induces responses by inhibiting adaptive immune resistance." *Nature* **515**(7528): 568-571.

Ueda, R. (2015). "Clinical Application of Anti-CCR4 Monoclonal Antibody." *Oncology* **89 Suppl 1**: 16-21.

Wang, Y., H. Wang, H. Yao, C. Li, J. Y. Fang and J. Xu (2018). "Regulation of PD-L1: Emerging Routes for Targeting Tumor Immune Evasion." *Front Pharmacol* **9**: 536.

Zamarin, D., J. M. Ricca, S. Sadekova, A. Oseledchik, Y. Yu, W. M. Blumenschein, J. Wong, M. Gigoux, T. Merghoub and J. D. Wolchok (2018). "PD-L1 in tumor microenvironment mediates resistance to oncolytic immunotherapy." *J Clin Invest* **128**(4): 1413-1428.

Reviewers' comments:

Reviewer #2 (Remarks to the Author):

Most of my comments have been addressed or discussed appropriately.
Tobias Bald

Reviewer #3 (Remarks to the Author):

Imbert et al. has responded all the comments raised by two reviewers regarding their revised manuscript. All the responses are appropriate for the data generated by the author groups. Although the experiments were sophisticated, the data generated in this manuscript are limited by relying heavily on one transplant mouse model and shRNA knockdown of SK1. In particular, the 2nd revised manuscript showed so dramatic differences between Day 11 and Day 20 shRNA knockdown. To truly resolve the concerns from previous reviewers, the best and also a simple approach would demonstrate the human relevance. It is necessary for the author group to obtain a cohort of Braf mutated human melanomas to address the human relevance. If the author group is able to conduct a prospective clinical trial, they should have the capacity of obtaining a retrospective cohort of melanomas with Braf mutation.

Response to referees

Reviewer #3

Imbert et al. has responded all the comments raised by two reviewers regarding their revised manuscript. All the responses are appropriate for the data generated by the author groups. Although the experiments were sophisticated, the data generated in this manuscript are limited by relying heavily on one transplant mouse model and shRNA knockdown of SK1. In particular, the 2nd revised manuscript showed so dramatic differences between Day 11 and Day 20 shRNA knockdown.

As stated by Reviewer #3, we did perform experiments using the mouse melanoma Yumm 1.7 cells (which are BRAF-mutated) transduced with control shRNAs or shRNA directed against SK1. Of importance, not a single one but 3 different shRNAs were used in these experiments. In addition, as requested by Reviewer #2, we performed rescue experiments to validate our concept (Supplementary Fig. 9). Furthermore, to complement the epigenetic strategy, we used a pharmacological approach and demonstrated that SK1 inhibition markedly enhanced the responses to anti-CTLA-4 and anti-PD-1 in murine models of both melanoma and colon cancer (new Fig. 5D-H). Our demonstration of the deleterious impact tumor SK1 plays on cancer development and ICI resistance was also validated in a breast cancer model using another genetic background (new Fig. 5A-C). Importantly, both colon and breast cancer models do not exhibit BRAF mutation. Thus, our data do not “rely on one model” but on different ones and by using different approaches. These latter data, initially described in Supplementary Figs. 7 and 8, have been moved in a new Figure 5.

Regarding the difference in the tumor immune infiltrate between Day 11 and Day 20, as requested by Reviewer #2, we performed additional experiments to analyse the frequencies of PD-1+ Eomes+ CD8+ T cells. We observed more exhausted CD8+ TILs (PD-1+ and Eomes+) at Day 20, irrespectively of the level of SK1 expression. Thus, this T cell exhaustion hypothesis cannot explain the regression of tumors already observed at day 11 upon SK1 silencing and ICI treatment. It is however not surprising to see such an increase in T cell exhaustion at Day 20: this has been vastly documented in the literature. Reversing exhaustion of T cells (a long-term process) does not seem to be the main mechanism through which SK1 inhibition promotes ICI efficacy at early stages, as opposed to the dramatic impact SK1 silencing exerts on the Treg population. Indeed, we still observed the increase of the CD8/Treg ratio in SK1-silenced tumors at Days 11, 20 and 23 (see Figure 1 below, not to be inserted in the manuscript).

Figure 1. CD8/Treg ratio in shCtrl and shSK1(1) tumors at day 11, 20 and 23.

In this study, our goal was not to dissect the role possibly played by SK1 from melanoma cells in T cell exhaustion (dysfunction). The main conclusion of our study is that “targeting SK1 decreases Treg accumulation and enhances ICI efficacy in mouse melanoma”. Our results (Fig. 2, 3 and 4 and Supplementary Fig. 2, 3 and 4) fully support this.

To truly resolve the concerns from previous reviewers, the best and also a simple approach would demonstrate the human relevance. It is necessary for the author group to obtain a cohort of Braf mutated human melanomas to address the human relevance. If the author group is able to conduct a prospective clinical trial, they should have the capacity of obtaining a retrospective cohort of melanomas with Braf mutation.

Firstly, we are really surprised to read this additional query (i.e., to use a cohort of BRAF mutated patients) as in your message, dated Sept. 9, 2019, you wrote use “We will let the new reviewer know that we are not looking to introduce additional hurdles for you at this round of review”.

Secondly, regarding the focus of our work, there is no scientific rationale to use BRAF-mutated patients. This at least for two reasons: (i) the most common therapy for BRAF-mutated melanoma patients is the combination of dabrafenib (BRAF inhibitor) plus trametinib (MEK inhibitor). Thus, it is not that easy to get access to melanoma biopsies from BRAF-mutated patients treated with ICI; (ii) BRAF mutation does not seem to be involved in the resistance to immunotherapy as demonstrated by several studies (Larkin et al. 2019; Wolchok et al. 2017; Wolchok, Rollin, and Larkin 2017).

Until very recently, the first line of treatment for BRAF-mutated melanoma patients used BRAF inhibitors in combination or not with MEK inhibitors. Since patients benefiting from ICI can display long-lasting remissions, BRAF-mutated melanoma patients displaying grade 3 disease now receive anti-PD-1 combined or not with anti-CTLA-4 as first line treatment (Eggermont et al. 2018; Long et al. 2017; Weber et al. 2017). Taking into account these recent changes in the treatment options for BRAF-mutated melanoma patients, retrospective tissue cohorts are still in the early stages of their building. This is why it is extremely difficult to access retrospective cohorts for such patients and why actually we are now performing one prospective study (Immusphinx:NCT03627026).

Finally, we believe this new request from the reviewers is technically impossible to fulfill in 3 months and is unlikely to yield a meaningful outcome.

References

Eggermont, A. M. M., C. U. Blank, M. Mandala, G. V. Long, V. Atkinson, S. Dalle, A. Haydon, M. Lichinitser, A. Khattak, M. S. Carlino, S. Sandhu, J. Larkin, S. Puig, P. A. Ascierto, P. Rutkowski, D. Schadendorf, R. Koornstra, L. Hernandez-Aya, M. Maio, A. J. M. van den Eertwegh, J. J. Grob, R. Gutzmer, R. Jamal, P.

- Lorigan, N. Ibrahim, S. Marreaud, A. C. J. van Akkooi, S. Suci, and C. Robert. 2018. 'Adjuvant Pembrolizumab versus Placebo in Resected Stage III Melanoma', *N Engl J Med*, 378: 1789-801.
- Larkin, J., V. Chiarion-Sileni, R. Gonzalez, J. J. Grob, P. Rutkowski, C. D. Lao, C. L. Cowey, D. Schadendorf, J. Wagstaff, R. Dummer, P. F. Ferrucci, M. Smylie, D. Hogg, A. Hill, I. Marquez-Rodas, J. Haanen, M. Guidoboni, M. Maio, P. Schoffski, M. S. Carlino, C. Lebbe, G. McArthur, P. A. Ascierto, G. A. Daniels, G. V. Long, L. Bastholt, J. I. Rizzo, A. Balogh, A. Moshyk, F. S. Hodi, and J. D. Wolchok. 2019. 'Five-Year Survival with Combined Nivolumab and Ipilimumab in Advanced Melanoma', *N Engl J Med*.
- Long, G. V., A. Hauschild, M. Santinami, V. Atkinson, M. Mandala, V. Chiarion-Sileni, J. Larkin, M. Nyakas, C. Dutriaux, A. Haydon, C. Robert, L. Mortier, J. Schachter, D. Schadendorf, T. Lesimple, R. Plummer, R. Ji, P. Zhang, B. Mookerjee, J. Legos, R. Kefford, R. Dummer, and J. M. Kirkwood. 2017. 'Adjuvant Dabrafenib plus Trametinib in Stage III BRAF-Mutated Melanoma', *N Engl J Med*, 377: 1813-23.
- Weber, J., M. Mandala, M. Del Vecchio, H. J. Gogas, A. M. Arance, C. L. Cowey, S. Dalle, M. Schenker, V. Chiarion-Sileni, I. Marquez-Rodas, J. J. Grob, M. O. Butler, M. R. Middleton, M. Maio, V. Atkinson, P. Queirolo, R. Gonzalez, R. R. Kudchadkar, M. Smylie, N. Meyer, L. Mortier, M. B. Atkins, G. V. Long, S. Bhatia, C. Lebbe, P. Rutkowski, K. Yokota, N. Yamazaki, T. M. Kim, V. de Pril, J. Sabater, A. Qureshi, J. Larkin, P. A. Ascierto, and Collaborators CheckMate. 2017. 'Adjuvant Nivolumab versus Ipilimumab in Resected Stage III or IV Melanoma', *N Engl J Med*, 377: 1824-35.
- Wolchok, J. D., V. Chiarion-Sileni, R. Gonzalez, P. Rutkowski, J. J. Grob, C. L. Cowey, C. D. Lao, J. Wagstaff, D. Schadendorf, P. F. Ferrucci, M. Smylie, R. Dummer, A. Hill, D. Hogg, J. Haanen, M. S. Carlino, O. Bechter, M. Maio, I. Marquez-Rodas, M. Guidoboni, G. McArthur, C. Lebbe, P. A. Ascierto, G. V. Long, J. Cebon, J. Sosman, M. A. Postow, M. K. Callahan, D. Walker, L. Rollin, R. Bhore, F. S. Hodi, and J. Larkin. 2017. 'Overall Survival with Combined Nivolumab and Ipilimumab in Advanced Melanoma', *N Engl J Med*, 377: 1345-56.
- Wolchok, J. D., L. Rollin, and J. Larkin. 2017. 'Nivolumab and Ipilimumab in Advanced Melanoma', *N Engl J Med*, 377: 2503-04.